_Article_

# Paladin is a phosphoinositide phosphatase regulating endosomal VEGFR2 signalling and angiogenesis

Anja Nitzsche[1,†,‡] iD, Riikka Pietilä[1,†], Dominic T Love[1,†], Chiara Testini[1,§], Takeshi Ninchoji[1], Ross O Smith[1], Elisabet Ekvärn[1,¶], Jimmy Larsson[1,††], Francis P Roche[1], Isabel Egaña[1], Suvi Jauhiainen[1], Philipp Berger[2], Lena Claesson-Welsh[1] iD & Mats Hellström[1,*] iD

## Abstract

Cell signalling governs cellular behaviour and is therefore subject to tight spatiotemporal regulation. Signalling output is modulated by specialized cell membranes and vesicles which contain unique combinations of lipids and proteins. The phosphatidylinositol 4,5-bisphosphate ($PI(4,5)P_2$), an important component of the plasma membrane as well as other subcellular membranes, is involved in multiple processes, including signalling. However, which enzymes control the turnover of non-plasma membrane $PI(4,5)P_2$, and their impact on cell signalling and function at the organismal level are unknown. Here, we identify Paladin as a vascular $PI(4,5)P_2$ phosphatase regulating VEGFR2 endosomal signalling and angiogenesis. Paladin is localized to endosomal and Golgi compartments and interacts with vascular endothelial growth factor receptor 2 (VEGFR2) _in vitro_ and _in vivo_. Loss of Paladin results in increased internalization of VEGFR2, over-activation of extracellular regulated kinase 1/2, and hypersprouting of endothelial cells in the developing retina of mice. These findings suggest that inhibition of Paladin, or other endosomal $PI(4,5)P_2$ phosphatases, could be exploited to modulate VEGFR2 signalling and angiogenesis, when direct and full inhibition of the receptor is undesirable.

**Keywords** endocytosis; Paladin; phosphatase; phosphoinositide; VEGFR2
**Subject Categories** Membranes & Trafficking; Vascular Biology & Angiogenesis

## Introduction

In the eukaryotic cell, membranes in different subcellular compartments play distinct roles in cell signalling. Growth factor receptor signalling is initiated at the cell surface and continues after internalization and during endosome trafficking (Lampugnani _et al_, 2006; Simons _et al_, 2016). However, signalling is quantitatively and qualitatively distinct depending on the specialized membrane compartment (Di Paolo & De Camilli, 2006). Key to the maintenance of membrane specialization are lipid kinases and phosphatases that phosphorylate/dephosphorylate distinct phospholipids with an inositol head group, i.e., phosphoinositides (PI). PIs are specifically distributed to generate "membrane codes" on intracellular vesicles and the plasma membrane (Di Paolo & De Camilli, 2006; Lemmon, 2008). These PIs together with Rab GTPases are required for the maintenance and coordination of endocytosis and membrane trafficking (Jean & Kiger, 2012) through recruitment of effector proteins to assemble specific endocytic complexes (Botelho _et al_, 2008; Jin _et al_, 2008; Lemmon, 2008; Chagpar _et al_, 2010; Mizuno-Yamasaki _et al_, 2010). Consequently, as lipid kinases and phosphatases are key regulators of membrane identity and function, they are also regulators of cell signalling. However, the kinases and phosphatases involved in the generation of the specific PIs at distinct subcellular localizations are still not fully identified and their roles at the organismal level are only partially known.

PIs can become phosphorylated at the 3′, 4′, and 5′ position of the inositol ring, giving rise to seven different PI species. The main PIs in the plasma membrane, early endosomes, late endosomes, and the Golgi apparatus are $PI(4,5)P_2$, $PI(3)P$, $PI(3,5)P_2$, and $PI(4)P$, respectively (Tan _et al_, 2015). These PI pools, present in microdomains of membrane vesicles, provide a unique environment for signalling and sorting (Tan _et al_, 2015).

1   Science for Life Laboratory, The Rudbeck Laboratory, Department of Immunology, Genetics and Pathology, Uppsala University, Uppsala, Sweden
2   Laboratory of Nanoscale Biology, Paul-Scherrer Institute, Villigen, Switzerland
    *Corresponding author. Tel: +46 708 717001; E-mail: mats.hellstrom@igp.uu.se
    †These authors contributed equally to this work
    ‡Present address: Université de Paris, Paris Cardiovascular Research Center, INSERM U970, Paris, France
    §Present address: Division of Nephrology, Department of Medicine, Boston Children's Hospital, Boston, MA, USA
    ¶Present address: Cepheid AB, Solna, Sweden
    ††Present address: Department of Cell and Molecular Biology, Uppsala University, Uppsala, Sweden

Growth factor signalling is initiated at the plasma membrane and involves activation of enzymes that use PIs as substrates. PI $(4,5)P_2$ at the plasma membrane is a substrate for phosphoinositide-3′ kinase (PI 3-kinase) resulting in generation of the second messenger $PI(3,4,5)P_3$, while hydrolysis of $PI(4,5)P_2$ by phospholipase C (PLC) generates inositol-1,4,5-trisphosphate and diacylglycerol (Katan & Cockcroft, 2020). Growth factor stimulation moreover leads to clathrin-mediated endocytosis whereby the PI $(4,5)P_2$ membrane pool is metabolized to $PI(3)P$ via $PI(4)P$ and $PI(3,4)P_2$ intermediates (He *et al*, 2017). $PI(4,5)P_2$ is also present to a lower extent in intracellular membranes, as demonstrated by immuno-electron microscopy and further suggested by the presence of lipid kinases and phosphatases for which $PI(4,5)P_2$ is a substrate or product. An important role for $PI(4,5)P_2$ dephosphorylation has been identified in growth factor receptor internalization and sorting in early endosomes. For example, PI $(4,5)P_2$ generated by type I gamma phosphatidylinositol phosphate 5-kinase i5 (PIPKIγi5) regulates sorting of endosomal epidermal growth factor receptor (EGFR). PIPKIγi5-deficiency results in reduced transition of the EGFR from endosomes to lysosomes and consequently prolonged signalling (Sun *et al*, 2013).

Paladin is a membrane-associated protein encoded by *Pald1* or *x99384/mKIAA1274* in mouse and *PALD1* or *KIAA1274* in human. Its expression is primarily restricted to endothelial cells during development (Wallgard *et al*, 2008; Suzuki *et al*, 2010; Wallgard *et al*, 2012). Although Paladin contains a phosphatase domain, it reportedly lacks protein phosphatase activity and was thus suggested to be a catalytically inactive pseudophosphatase (Huang *et al*, 2009; Roffers-Agarwal *et al*, 2012; Kharitidi *et al*, 2014; Reiterer *et al*, 2014). However, Paladin has been implicated in various cell signalling pathways. A broad phenotypic screen in *Pald1* null mice covering all organ systems revealed a specific lung phenotype, i.e., an emphysema-like lung histology and increased turnover of lung endothelial cells (Egana *et al*, 2017). In addition, studies on chick embryos support a role for Paladin in neural crest migration (Roffers-Agarwal *et al*, 2012). Cell culture studies suggest that Paladin negatively regulates expression and phosphorylation of the insulin receptor, as well as the phosphorylation of the downstream serine/threonine kinase AKT (Huang *et al*, 2009). Furthermore, Paladin is a negative regulator of Toll-like receptor 9 (TLR9) signalling (Li *et al*, 2011). Collectively, these observations suggest that Paladin is an important player in cell signalling. Nevertheless, the mechanism whereby Paladin achieves those effects on diverse signalling pathways has remained unknown.

Here, we provide evidence that Paladin is a $PI(4,5)P_2$ phosphatase that lacks phospho-tyrosine/serine/threonine phosphatase activity. Paladin localized to endosomal vesicles where it interacted with VEGFR2, thereby positioned as a potential regulator of endosomal trafficking. In line with this, loss of *Pald1* expression led to faster VEGFR2 internalization to $EEA1^+$ endosomes and increased pERK1/2 levels *in vitro* and *in vivo* after VEGF-A stimulation. Phenotypically, *Pald1* deficiency promoted enhanced pathological retinal angiogenesis. *Pald1* deficiency also resulted in retinal vascular hypersprouting, which was normalized by inhibition of MEK.

## Results and Discussion

### Paladin is an endosomal PI(4,5)P2 phosphatase interacting with VEGFR2

Despite the lack of published experimental evidence, Paladin had been postulated to be a catalytically inactive pseudophosphatase (Huang *et al*, 2009; Kharitidi *et al*, 2014; Reiterer *et al*, 2014). However, more recently, Alonso and Pulido suggested that Paladin is a Cys-based phosphatase, which forms its own subclass (IV). Whereas all the neighbouring phosphatase subclasses dephosphorylate PI, they proposed, based on structural similarity, that Paladin might possess inositol phosphatase activity (Alonso & Pulido, 2015). The Paladin amino acid sequence contains four repeats of the minimal protein tyrosine phosphatases (PTP) consensus sequence $CX_5R$ (Fig EV1A) (Wallgard *et al*, 2012). Two of these repeats share high similarity with the extended conserved signature motif of the PTP active site, but importantly Paladin lacks the conserved histidine residue preceding the $CX_5R$ motif (Fig EV1B) (Andersen *et al*, 2001). However, an increasing number of PTPs are shown to have phosphoinositides as substrates (Pulido *et al*, 2013) and several new candidate PI phosphatases have been proposed, including Paladin (Alonso & Pulido, 2015). We therefore used a colorimetric screen based on the release of free phosphate to evaluate such phosphatase activity of Paladin. We expressed and immunoprecipitated V5-tagged Paladin and the phosphatase and tensin homolog (PTEN) in HEK293 cells. Wild-type PTEN and dephosphorylation of $PI(3,4,5)P_3$ was used as positive control and the C124S phosphatase-dead PTEN variant as negative control. Similarly, we used a Paladin variant with a cysteine to serine (C/S) substitution of all four cysteines in the $CX_5R$ motifs as a negative control (Fig EV1A). Indeed, wild-type Paladin showed specific phosphatase activity towards $PI(4,5)P_2$ and tended to also dephosphorylate $PI(3,4,5)P_3$ but not PI monophosphates or inositol phosphates (Figs 1A and EV1C). Further, using a radioactively labelled phosphopeptide substrate and the protein tyrosine phosphatase, T cell (TC)-PTP, as a positive control we confirmed the data by Huang and co-workers that Paladin lacks phospho-tyrosine activity (Fig EV1D; Huang *et al*, 2009). Similarly, no phosphatase activity against a protein kinase C (PKC)-phosphorylated phosphoserine/phosphothreonine peptide was apparent (Fig EV1E). These observations support the conclusion that Paladin is a phosphoinositide phosphatase.

Paladin is preferentially expressed in endothelial cells during development (Wallgard *et al*, 2012). Accordingly, we used immunostaining to evaluate the subcellular localization of Paladin in primary human dermal microvascular endothelial cells (HDMEC). The analysis revealed a vesicular staining pattern of Paladin with enrichment in the perinuclear region overlapping with Golgi staining, but not with the plasma membrane identified by immunostaining of the junctional protein vascular endothelial (VE) cadherin (Figs 1B and EV1F). However, we observed Paladin co-localization with VEGFR2-positive vesicles which was increased after VEGF-A stimulation (Fig 1C and D). Further, Proximity Ligation Assay (PLA) was used to assess a possible interaction between Paladin and VEGFR2 over time after VEGF-A stimulation. A low level of proximity between Paladin and VEGFR2 existed in the basal state that rapidly increased after 2 min of VEGF-A stimulation and was still

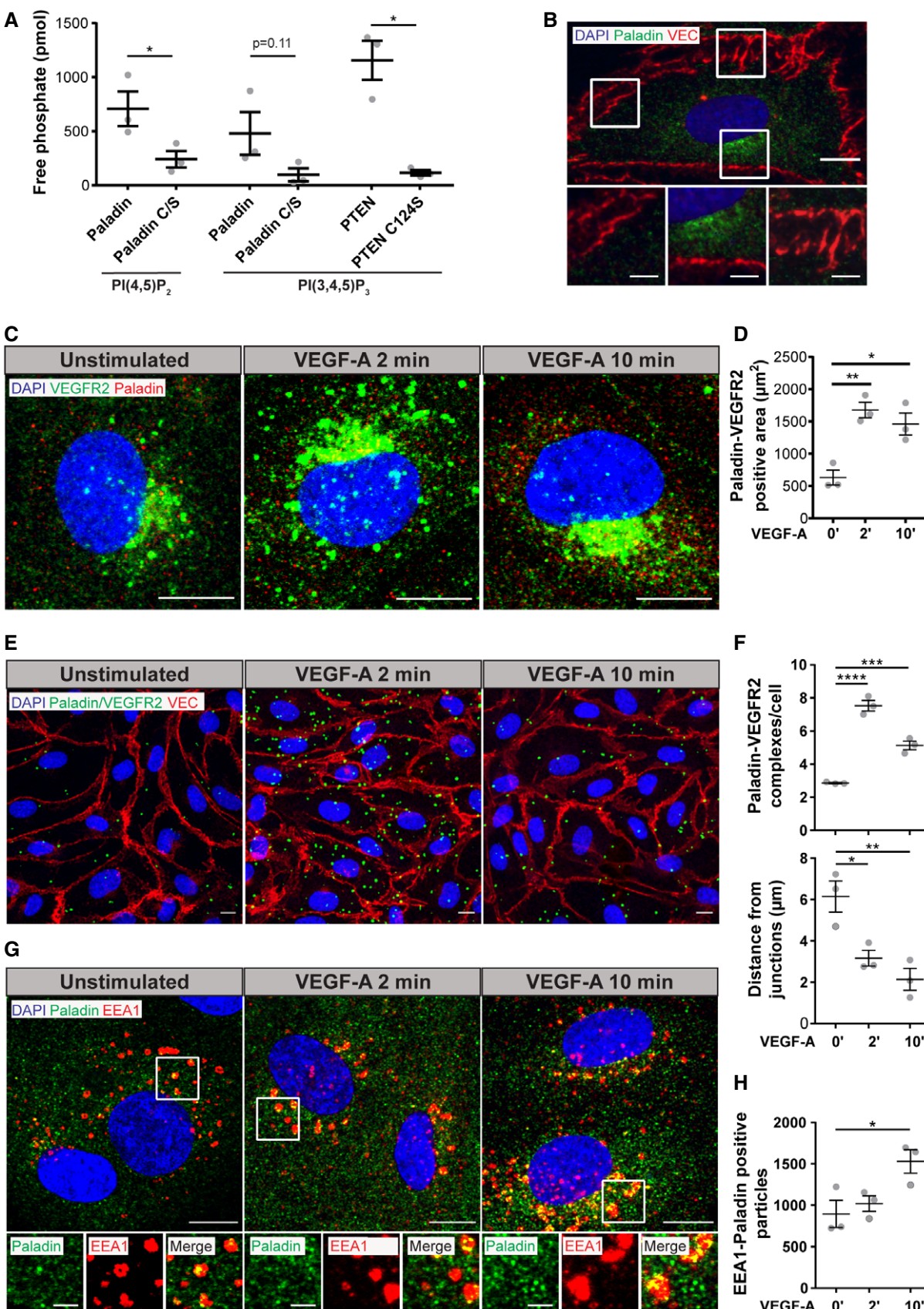

**Figure 1.**

**Figure 1. Paladin is a lipid phosphatase.**

A  Lipid phosphatase activity of Paladin, wild-type and phosphatase-dead C/S mutant, toward $PI(4,5)P_2$ and $PI(3,4,5)P_3$ substrates. Positive control; wild-type phosphatase and tensin homolog (PTEN); negative control; lipid phosphatase-dead C124S PTEN. Mean $\pm$ SEM, Paired $t$-test, $n = 3$ biological replicates.

B  Representative confocal image of HDMEC stained for Paladin (green), VE Cadherin (VEC, red), and nuclei (DAPI, blue), scale bar: 10 μm. White boxes in the upper image are showed at higher magnification below, scale bar: 3 μm.

C  HDMEC stained for VEGFR2 (green), Paladin (red), and nuclei (DAPI, blue) and stimulated with 50 ng/ml VEGF-A for 0, 2, or 10 min. Scale bar: 10 μm.

D  Quantification of VEGFR2-Paladin co-staining before and after VEGF-A stimulation as shown in (C). Mean $\pm$ SEM, one-way ANOVA. $n = 3$ biological replicates.

E  HDMEC analysed using Proximity Ligand Assay (PLA) for Paladin and VEGFR2. Green dots indicate complex formation, VE cadherin (red), and nuclei (DAPI, blue). Cells stimulated with 50 ng/ml VEGF-A for 0, 2, or 10 min. Scale bar: 10 μm.

F  Quantification of (E), the number of PLA Paladin-VEGFR2 complexes per cell at 0, 2, and 10 min after VEGF-A stimulation (top). Quantification of (E), average distance for the PLA complexes to the nearest VE cadherin positive junction at 0, 2, and 10 min after VEGF-A stimulation (bottom). Mean $\pm$ SEM, one-way ANOVA. $n = 3$ biological replicates.

G  HDMEC stained for Paladin (green) and EEA1 (red), colocalization in yellow, and nuclei (DAPI, blue) after VEGF-A stimulation for 0, 2, or 10 min, a representative image from a single confocal plane is shown, scale bar: 10 μm. White boxes in the upper image are showed at higher magnification below, scale bar 3 μm. See Appendix Fig S1 for siRNA *PALD1* knockdown controls.

H  Quantification of (G), EEA1/Paladin double-positive particles per field of view at 0, 2, and 10 min after VEGF-A stimulation. Mean $\pm$ SEM, one-way ANOVA. $n = 3$ biological replicates.

Data information: *$P < 0.05$, **$P < 0.01$, ***$P < 0.001$, ****$P < 0.0001$.

maintained, but at a lower level, at 10 min (Figs 1E and F, and EV1G). The VEGF-A-induced complexes appeared close to the cell border, compared with basal complexes (Fig 1E and F). VEGFR2 immunoprecipitation confirmed complex formation with Paladin *in vitro* (in primary endothelial cells) and *in vivo* (in mouse), but the interaction required blocking of dephosphorylation by peroxy-vanadate treatment (Fig EV1H and I). Given the early induction of complex formation between Paladin and VEGFR2 after VEGF-A treatment, we explored the relationship between Paladin and the early endosome antigen 1 (EEA1). Paladin decorated microdomains of EEA1[+] vesicles and the number of vesicles positive for both EEA1 and Paladin increased after VEGF-A stimulation (Fig 1G and H, and Appendix Fig S1).

Taken together, Paladin catalyses $PI(4,5)P_2$ dephosphorylation and is present in endosomal vesicles while quickly depleting towards the cell periphery. Paladin appears in close proximity to VEGFR2 in response to VEGF-A stimulation.

**Paladin regulates VEGFR2 internalization and early endosomal trafficking**

To test whether Paladin affects VEGFR2 trafficking and signalling, the effect of siRNA-mediated knockdown of *PALD1* in HDMEC was analysed. *PALD1* siRNA treatment resulted in a marked, 35–51% increase of the total basal VEGFR2 pool (Figs 2A and C, and EV2A). However, the receptor was degraded similarly over time after VEGF-A stimulation when comparing *PALD1* siRNA and control-treated cells (Figs 2A and EV2B). To study the effect of the presence and absence of Paladin on the trafficking of surface VEGFR2, endothelial cells, in which *PALD1* expression had been silenced or not, were treated with VEGF-A for different time periods. Cell surface biotinylation after VEGF-A stimulation was used to pull down VEGFR2 by streptavidin beads, separating the cell surface-localized VEGFR2 pool from the internal pool. When normalized to total VEGFR2 levels, the amount of VEGFR2 at the cell surface in control and *PALD1* siRNA-treated cells was not significantly different after VEGF-A treatment (Figs 2A and B, and EV2C). In a parallel analysis, we evaluated the size of the internalized VEGFR2 pool over time, by cell surface biotinylation prior to VEGF-A stimulation and subsequent stripping of remaining cell surface biotin, allowing the pull down of only protected, internalized proteins. After 15-min treatment with VEGF-A and normalization to total VEGFR2, the internalized VEGFR2 pool in *PALD1*-silenced endothelial cells was almost twice that of the control culture (Figs 2C and D, and EV2D). This suggests that Paladin controls the rate of VEGFR2 internalization at the early time points after VEGF-A stimulation.

To track the early localization of VEGFR2 after internalization, we stained *PALD1* siRNA-treated cells for VEGFR2 and the early endosome marker EEA1. Already after 2 min of VEGF-A stimulation, there was a significant increase in EEA1[+] vesicle number as well as EEA1/VEGFR2 double-positive structures in the *PALD1* knockdown cells compared with siRNA control; the number of double-positive vesicles further increased with 10 min VEGF-A treatment (Fig 2E and F).

To visualize the phosphoinositide substrate in intact cells expressing Paladin or not, we stained *PALD1* siRNA-treated cells for $PI(4,5)P_2$ after VEGF-A treatment. $PI(4,5)P_2$ was localized along the plasma membrane as well as in vesicles throughout the cells in the basal condition. After VEGF-A stimulation, there was an increase in the vesicular $PI(4,5)P_2$ signal over time. In the *PALD1* knockdown cells, there was a prominent increase of $PI(4,5)P_2$ signal at 2 min after VEGF-A stimulation, as compared to control siRNA-treated cells (Fig 2G and H). This is compatible with the *in vitro* data that $PI(4,5)P_2$ is a substrate for Paladin.

Given the timing of maximal VEGFR2-Paladin interaction and increase of VEGFR2 internalization, we suggest that Paladin controls the early steps of endosomal trafficking. This is in line with the early and prominent increase of $PI(4,5)P_2$ in cells lacking Paladin as it is known that $PI(4,5)P_2$ from the plasma membrane is quickly dephosphorylated after clathrin mediated endocytosis (He *et al*, 2017). However, other roles for Paladin in the endosomal compartment cannot be ruled out, e.g., the role of $PI(4,5)P_2$ in late endosomes in controlling sorting of recycling endosomes (Tan *et al*, 2015). Further studies are needed to determine the exact, and potentially multiple, role(s) of Paladin in different stages of endosomal trafficking. Moreover, an effect of Paladin on early stages of internalization is likely to have an impact on later stages of trafficking.

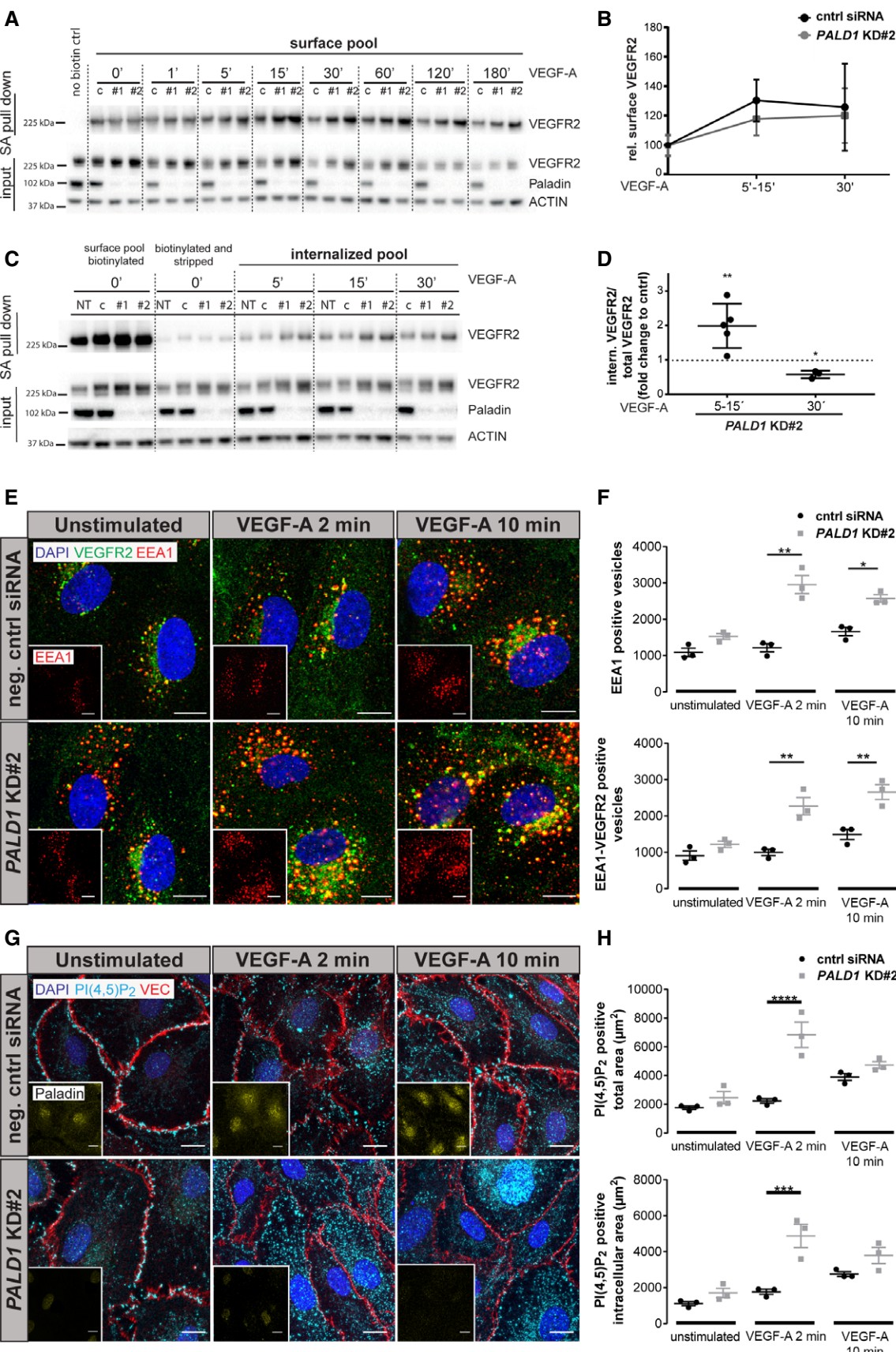

**Figure 2.**

◄

**Figure 2.  Paladin regulates VEGFR2 internalization and endosomal trafficking.**

A   Cell surface VEGFR2 levels detected by cell surface biotinylation, using thiol-cleavable sulfo-NHS-SS-biotin, of HDMEC transfected with *PALD1* siRNA (#1 and #2) or non-targeting control ("c") siRNA, followed by VEGF-A stimulation (50 ng/ml) for indicated time periods. Total lysates (input) and streptavidin (SA) pull down, immunoblotted for VEGFR2, Paladin, and actin. 'No biotin ctrl', cells not treated with sulfo-NHS-SS-biotin.

B   Quantification of data in (A). VEGFR2 surface levels (data pooled for the indicated time points) normalized to total VEGFR2 levels and compared between control and siRNA-treated HDMEC. $n = 4$ for each time point, biological replicates, Mean $\pm$ SEM.

C   Internalized pool of VEGFR2 after VEGF-A treatment (50 ng/ml) of non-transfected HDMEC ("NT") or HDMEC transfected with *PALD1* siRNA (#1 and #2) or non-targeting control siRNA ("c"). Cell surface biotinylation was performed prior to VEGF-A stimulation and at indicated time points, remaining cell surface biotin was stripped and the internalized pool of VEGFR2 was collected by SA pull down. Immunoblotting of the total lysate (input) and SA pull down fraction for VEGFR2, Paladin, and actin.

D   Quantification of data in (C). Data were normalized to total VEGFR2 levels in the lysate after subtraction of signals in biotinylated and stripped samples. Mean $\pm$ SEM, unpaired *t*-test for indicated time points, normalized to control siRNA sample. $n = 3$ for each time point, biological replicates.

E   Analysis of EEA1 and VEGFR2 vesicles following *PALD1* knockdown. Representative images of VEGFR2 (green)/EEA1 (red) double-positive (yellow) vesicles in negative control siRNA, and *PALD1* KD#2 siRNA-silenced HDMEC at 0, 2, and 10 min of VEGF-A stimulation (50 ng/ml). DAPI in blue, scale bar: 10 μm. Inset shows only EEA1 channel, scale bar: 10 μm.

F   Quantification of (E), number of EEA1 positive (top) or VEGFR2-EEA1 double-positive vesicles (bottom) per field of view. Mean $\pm$ SEM, two-way ANOVA, $n = 3$ biological replicates.

G   HDMEC stained for PI(4,5)P$_2$ (cyan), VE cadherin (red), and Paladin (yellow in inset) following treatment using negative control or *PALD1* (KD#2) siRNA. VEGF-A stimulation for 0, 2, or 10 min (50 ng/ml). DAPI in blue. Scale bar: 10 μm.

H   Quantification of (G), Total PI(4,5)P$_2$ signal (top), or intracellular PI(4,5)P$_2$ not overlapping with VE cadherin (bottom). Mean $\pm$ SEM, two-way ANOVA, $n = 3$ biological replicates.

Data information: *$P < 0.05$, **$P < 0.01$, ***$P < 0.001$, ****$P < 0.0001$.
Source data are available online for this figure.

## Loss of Paladin leads to increased ERK1/2 phosphorylation downstream of VEGFR2

Having shown that Paladin regulates early VEGFR2 trafficking in response to VEGF-A, we then explored whether signalling downstream of VEGFR2 was affected by *PALD1* silencing. Phosphorylation of VEGFR2 normalized to total VEGFR2 was enhanced in *PALD1* siRNA-treated cells compared against non-silenced cells in response to VEGF-A (Fig 3A and B) in keeping with the notion that the more rapid internalization of VEGFR2 in the absence of Paladin protected against dephosphorylation by plasma membrane-associated protein tyrosine phosphatases (Lanahan *et al*, 2010). Furthermore, the phosphorylation of certain downstream targets of VEGFR2: mitogen-activated protein kinase (MAPK)3/MAPK1 (ERK1/2) and SRC were increased in *PALD1* silenced HDMEC (Figs 3C and D, and EV3A). In contrast, *PALD1* silencing did not affect the degree of phosphorylation of AKT downstream of VEGF-A/VEGFR2 (Fig EV3B). These observations indicated that certain pathway downstream of VEGFR2 were hyperactivated in the absence of Paladin in HDMEC.

We next investigated the loss of *Pald1* on VEGFR2 signalling *in vivo* using a global constitutive *Pald1* knockout mouse (Wallgard *et al*, 2012). We examined murine cardiac endothelial cells as an example of microcirculatory endothelial cells responsive to VEGF-A, moreover, heart endothelial cells express *Pald1* and, in contrast to lung endothelial cells, they do not show an overt phenotype in the *Pald1$^{-/-}$* mouse (Wallgard *et al*, 2012; Egana *et al*, 2017; Schaum *et al*, 2018). Accordingly, VEGF-A was injected into the tail vein of adult mice followed by retrieval and lysis of hearts at specific time points and immunoblotting. While we did not observe significant differences in VEGFR2 phosphorylation (Fig 3E and F), degradation of VEGFR2 was transiently delayed in *Pald1$^{-/-}$* heart lysates compared with the wild-type. The delay was manifested at 5–10 min after VEGF-A stimulation, while at 15–20 min, VEGFR2 levels were equivalent between the genotypes (Fig 3E and F). Furthermore, VEGF-A downstream signalling in the *Pald1$^{-/-}$* hearts was altered. After VEGF-A stimulation, Erk1/2 (Mapk3/Mapk1)

phosphorylation was increased and prolonged in *Pald1$^{-/-}$* mice compared with their wild-type littermates (Fig 3E and F). However, we did not observe statistically significant differences in the level of phosphorylation of phospholipase (PLC) γ, Akt, or Src in response to VEGF-A *in vivo* in *Pald1$^{-/-}$* compared with wild-type littermates (Figs 3E and F, and EV3C and D).

Taken together, loss of Paladin function *in vitro* and *in vivo* results in altered activation and degradation of VEGFR2, and increased Erk1/2 signalling downstream of VEGFR2. In contrast, GIPC/synectin deficiency is accompanied by prolonged VEGFR2 trafficking through early endosomes which allows for efficient VEGFR2 dephosphorylation by the tyrosine-protein phosphatase non-receptor type 1 (PTP1b) and consequently, reduced pERK1/2 levels (Lanahan *et al*, 2010). These data suggest the scenario that in *PALD1* deficiency, VEGFR2 can escape from dephosphorylation by PTP1b and thereby promote increased pERK1/2 levels.

## Endothelial hypersprouting in the *Pald1*-deficient postnatal retina results from exaggerated Erk1/2 signalling

Retinal endothelial cells proliferate and migrate in a VEGF-A/VEGFR2-dependent and highly stereotyped manner during early postnatal development (Gerhardt *et al*, 2003). We therefore investigated the consequence of *Pald1* gene inactivation on blood vessel development of early postnatal retinas in mice. We observed a reduced vascular outgrowth in the *Pald1$^{-/-}$* retina at postnatal day (P) 5 (Fig 4A and B) as well as an increase in the number of filopodia extensions from endothelial tip cells (Fig 4C and D). In addition, the density of the vascular front in the *Pald1$^{-/-}$* P5 retina was greater both in the capillary bed and around veins, as compared to the littermate control retina (Figs 4E and F, and EV4A). In line with this, endothelial cell sprouting from 3D spheroids was enhanced in *PALD1* siRNA-treated cells, compared with control siRNA (Fig EV4B). Further, HDMEC proliferation was enhanced in *PALD1* knockdown cells (Fig EV4C). These observations indicate that Paladin is a negative regulator of endothelial proliferation and angiogenic sprouting. In agreement with an increased pErk1/2

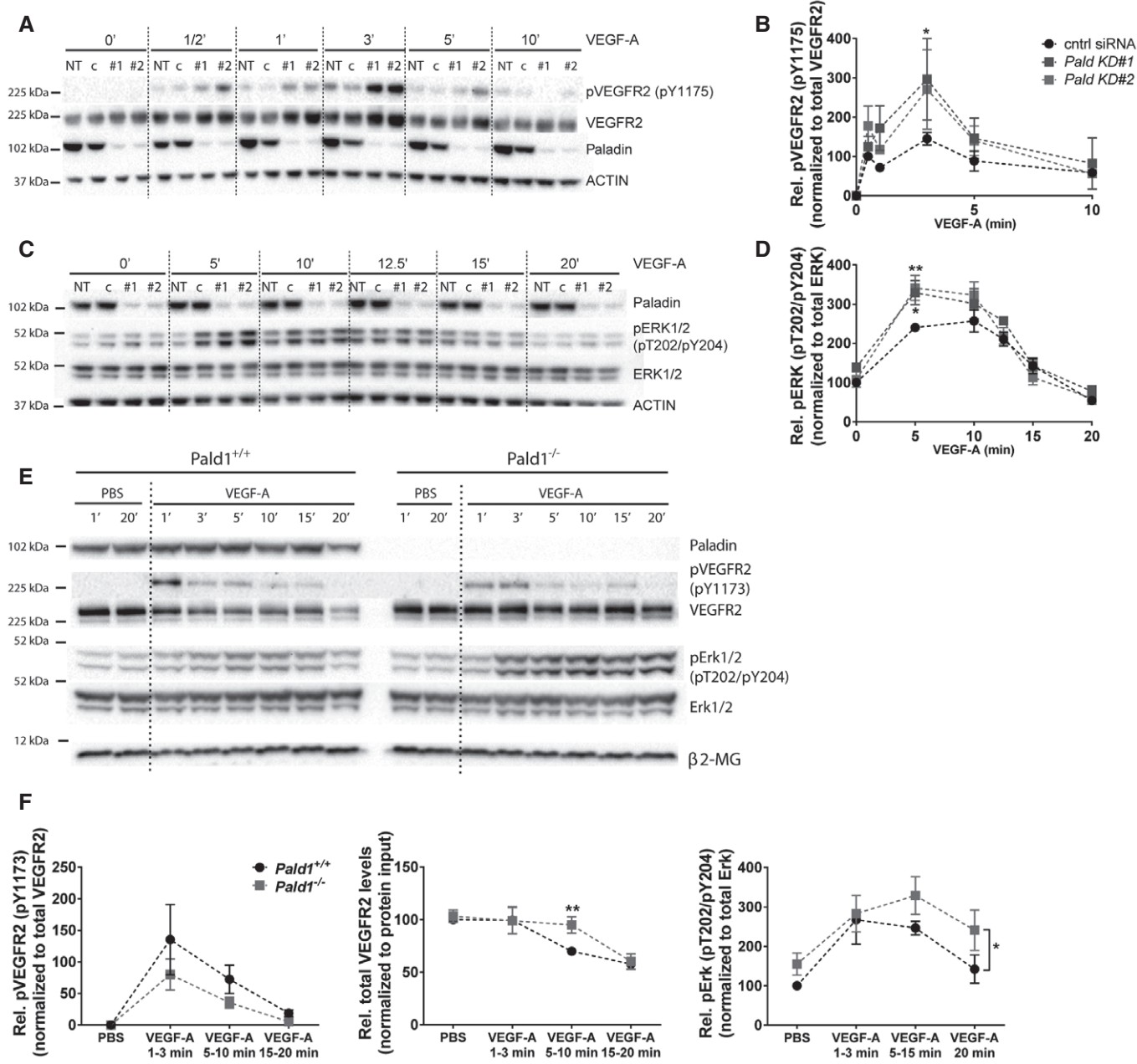

**Figure 3. Paladin regulates VEGFR2 signalling.**

A–D   Signalling downstream of VEGF-A/VEGFR2 assessed in HDMEC, untreated (NT), transfected with non-targeting siRNA (c/cntrl), or with two different *PALD1*-targeting siRNAs (KD#1 and KD#2), and treated with VEGF-A for 0–10 and 0–20 min, respectively. Immunoblotting of cell lysates for phosphorylated (p) VEGFR2 (pY1175), total VEGFR2, phosphorylated Erk1/2 (pT202 and pY204), and total Erk. Actin served as loading control (A, C). *PALD1* knockdown was verified by blotting for Paladin. Quantification of pVEGFR2 (normalized to total VEGFR2) (B) and pErk1/2 (normalized to total Erk1/2) (D). Mean ± SEM, two-way ANOVA, n = 3 biological replicates.

E, F   Immunoblotting of total heart lysates from adult *Pald1*[+/+] and *Pald1*[−/−] mice, tail vein injected with VEGF-A (0.25 µg/g body weight) or PBS for the indicated time points, for Paladin, phosphorylated, and total levels of VEGFR2, Erk1/2, and β2-microglobulin (β2-MG, loading control). (F) Quantification of pVEGFR pY1173 normalized to total VEGFR2 (n = 3), total VEGFR2 levels normalized to total loading control (n = 4), pT202/pY204 Erk1/2 normalized to Erk1/2 (n = 5). Mean ± SEM, multiple *t*-test with Holm-Sidak correction (total VEGFR2), two-way ANOVA (others).[‡‡]

Data information: *P < 0.05, **P < 0.01.
Source data are available online for this figure.

---

[‡‡]Correction added on 3 February 2021, after first online publication: the spelling of normalized has been corrected in the Y-axis title of the left-hand graph in Fig 3F.

accumulation in response to VEGF-A stimulation, as detected by immunoblotting of $Pald1^{-/-}$ hearts (Fig 3E and F), the area of vascular Erk1/2 immunostaining was increased in the $Pald1^{-/-}$ retina (Fig 4G and H). Erk1/2 activity regulates cyclin D1 (*Ccnd1*); in agreement, *Ccnd1* transcript levels increased in $Pald1^{-/-}$ retina

compared with littermate controls (Figs 4I and EV4D) and cyclin D1 staining trended to a greater number of positive nuclei in $Pald1^{-/-}$ compared with $Pald1^{+/+}$ (Fig 4J and K). To confirm that the observed phenotypes in the $Pald1^{-/-}$ retina were due to increased Erk1/2 signalling, an inhibitor of MAP2K1 (MEK), a dual Ser/Thr/

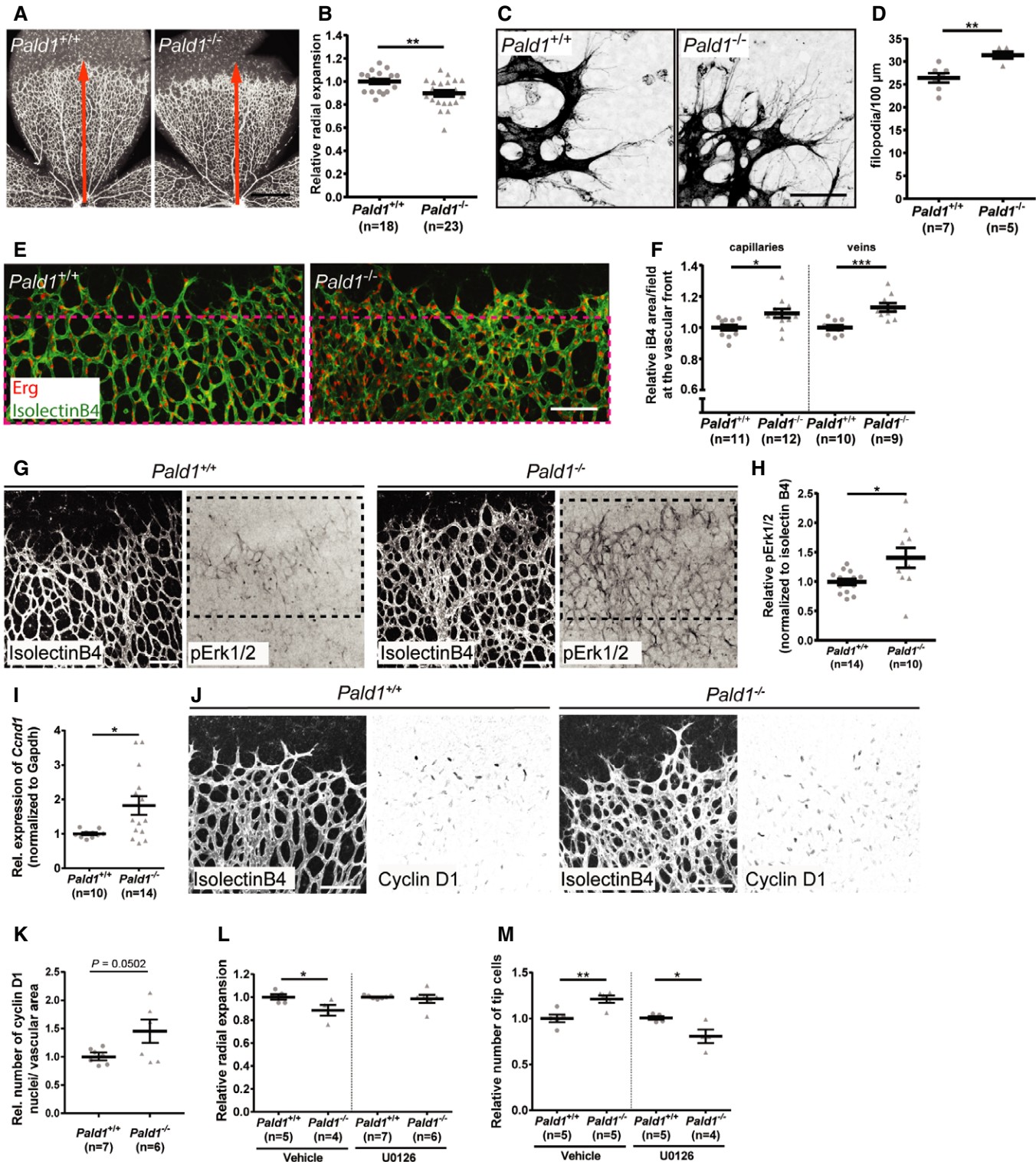

**Figure 4.**

◄

**Figure 4. Retinal vascular phenotype in Pald1⁻/⁻ mouse.**

A, B Delayed vascular outgrowth and hyperdense vascular front in isolectinB4-stained P5 retina from $Pald1^{-/-}$ mouse compared with $Pald1^{+/+}$ (A). Orange arrow indicates radial expansion of the vascular plexus in the $Pald1^{+/+}$ retina as a reference. Scale bar: 1 mm. Quantification of radial expansion (B) as shown in (A) normalized to wild-type litter mates. Mean ± SEM, unpaired t-test. n = 14 litters, 18 wild type, and 23 knockout pups.

C, D Increased filopodia number in $Pald1^{-/-}$ mouse retina at vascular front, visualized by isolectinB4 staining (C). Scale bar: 50 μm. Quantification of filopodia per 100 μm at the vascular front (D). Mean ± SEM, unpaired t-test. n = 3 litters, 7 wild type, and 5 knockout pups.

E, F P5 retina vascular front (isolectinB4, green) and endothelial nuclei (Erg, red) (E). Scale bar: 100 μm. Magenta stippled square area quantified in (F). Vascular density was determined in the capillary bed (left) (11 wild type, 12 knockout) and in area around veins (right) (10 wild type, 9 knockout, biological replicates). Mean ± SEM, unpaired t-test.

G, H IsolectinB4 (white) visualizes the entire vasculature. pT202/pY204 Erk1/2 immunostaining (black) in $Pald1^{+/+}$ and $Pald1^{-/-}$ P5 pups. Scale bar: 100 μm. Quantification of pT202/pY204 Erk1/2 area (H) as in the black stippled square in (G) (400 μm from the retina rim), normalized to isolectinB4 area. Mean ± SEM, unpaired t-test. n = 5 litters, 14 wild type, and 10 knockout retinas per genotype.

I Quantitative real-time PCR analysis of P4-P5 retinas from $Pald1^{+/+}$ and $Pald1^{-/-}$ pups. Ccnd1 transcript levels, normalized to Gapdh. Mean ± SEM, unpaired t-test. n = 10 $Pald1^{+/+}$ and 14 $Pald1^{-/-}$ pups.

J IsolectinB4 (white) and Cyclin D1 (black) staining of P5 retinas from $Pald1^{+/+}$ and $Pald1^{-/-}$ pups. Scale bar: 100 μm.

K Quantification of the number of Cyclin D1-positive nuclei normalized for isolectinB4 area in retinas from $Pald1^{+/+}$ and $Pald1^{-/}$ P5 pups. Mean ± SEM, unpaired t-test. n = 7 $Pald1^{+/+}$ and 6 $Pald1^{-/-}$ pups.

L Quantification of relative radial expansion in vehicle (n = 2 litters, 5 $Pald1^{+/+}$ and 4 $Pald1^{-/-}$ pups) and MEK inhibitor (U0126)-treated pups (n = 4 litters, 7 $Pald1^{+/+}$ and 5 $Pald1^{-/-}$ pups). MEK inhibitor/vehicle was administered twice at 12-h interval at P4 and eyes collected at P5. Each dot is one mouse. Mean ± SEM, one-way ANOVA, n = 4–7.

M Quantification of the tip cell number in vehicle- (n = 3 litters, 5 $Pald1^{+/+}$ and 5 $Pald1^{-/-}$ pups) and MEK inhibitor (U0126)-treated pups (n = 3 litters, 5 $Pald1^{+/+}$ and 4 $Pald1^{-/-}$ pups). MEK inhibitor/vehicle administered twice at P5 at 2-h intervals, and eyes collected 2 h after the second injection. Each dot is one mouse. Mean ± SEM, one-way ANOVA.

Data information: *$P < 0.05$, **$P < 0.01$, ***$P < 0.001$.

Tyr kinase upstream of Erk1/2 was employed. Notably, Erk1/2 phosphorylation in the P5 retina was reduced in a time-dependent manner after a single intraperitoneal dose with the MEK inhibitor U0126 (Fig EV4E). Treatment with U0126 normalized the vascular outgrowth and endothelial tip cell numbers in $Pald1^{-/-}$ retinas, which underscored the contribution of the exaggerated Erk1/2 signalling to this phenotype (Figs 4L and M, and EV4F and G). Collectively, these observations indicate that Paladin is a negative regulator of Erk1/2 signalling that in turn controls endothelial proliferation and sprouting in the early postnatal mouse retina. In agreement, Erk1/2 is an important and direct mediator of angiogenic sprouting in zebrafish, as it is both necessary and sufficient for endothelial tip cell sprouting (Shin et al, 2016). However, the context is likely more complex in the $Pald1^{-/-}$ retina, as defective endosomal trafficking not only affects VEGFR2 but most likely also other signalling molecules. Nevertheless, we observed normalization of the increased endothelial tip cell numbers and vascular outgrowth defects in the Pald1 knockout retina upon MEK inhibitor treatment, underscoring the role of Erk1/2 activation as an important part of the signalling defects in Pald1-deficient endothelial cells.

## Absence of Pald1 leads to increased pathological retinal angiogenesis

Pathological retinal angiogenesis is induced in hypoxia and VEGF-A is a known driver of the pathology in diseases such as wet age-related macular degeneration, where VEGF-A blockade is an important treatment (Mitchell, 2011). Since we identified a role for Paladin in regulating endothelial sprouting and VEGF-A/VEGFR2 signalling, the importance of Pald1 in pathological retinal angiogenesis was explored using an oxygen-induced retinopathy (OIR) model in mice to trigger vaso-obliteration and compensatory pathological angiogenesis (Connor et al, 2009). The $Pald1^{+/LacZ}$ mouse was employed to track activation of Pald1 transcription which showed endothelial LacZ expression in the retinal vasculature and in pathological blood vessels in the $Pald1^{+/LacZ}$ retina following OIR

(Fig 5A). Indeed, VEGF-A, but not other endothelial cell growth factors such as fibroblast growth factor-2 (FGF2) or stromal derived factor 1a (SDF1a) significantly induced expression of Paladin in endothelial cells in vitro (Figs 5B and EV5A). Moreover, Paladin was induced by VEGF-A in the retinal vasculature in vivo, as indicated by LacZ reporter expression (Fig 5C). In addition, mice lacking Pald1 exhibited increased vascular tuft formation at P17, but showed no difference in avascular area, compared with wild-type mice following OIR (Figs 5D–F and EV5B). Of note, we did not observe any differences in the vascular leakage in wild type and $Pald1^{-/-}$ after OIR, based on microsphere extravasation, or in the phosphorylation of VE cadherin at Y685, which correlates with VEGF-A-induced leakage (Smith et al, 2020), in the vascular tufts at P17 (Fig EV5C–F). In addition, as observed at the early developmental stage, pErk1/2 immunostaining intensity was increased in the vasculature at P15 in $Pald1^{-/-}$ retinas compared with $Pald1^{+/+}$ challenged in the OIR model (Fig 5G and H).

Taken together, Paladin is upregulated by VEGF-A, which is the main driver of pathological retinal angiogenesis in mouse and human, and in the OIR model Paladin functions as a negative regulator of Erk1/2 signalling and pathological angiogenesis. Importantly, PALD1 has been genetically associated with Moyamoya disease in two families. Moyamoya disease is caused by the occlusion of the carotid artery and its branches, causing characteristic pronounced collateral vessel formation and stroke in the central nervous system (Grangeon et al, 2019). A potential causal link between the hyperactive endothelial signalling observed in the $Pald1^{-/-}$ mouse and excessive collateral formation in patients with Moyamoya disease need further investigations.

Based on the evidence presented herein, we propose that Paladin is a critical regulator of VEGFR2 endosomal trafficking. The differences observed between in vitro tissue culture and in vivo mouse tissue models under conditions of Paladin deficiency, such as the total expression levels of VEGFR2 and levels of pVEGFR2 (upregulated in vitro but not in vivo), the rate of VEGFR2 degradation in response to VEGF-A (not affected in vitro but delayed in vivo) may

reflect differences between these models in the net effect of Paladin's presence/absence on VEGFR2 intracellular fate including recycling. However, that Paladin deficiency promotes Erk1/2 overactivity was clearly established *in vitro* as well as *in vivo*,

during development and in pathology. That Erk1/2 overactivity was a consequence of VEGF-A signalling *in vivo* was illustrated by the strong connection between the developmental and pathological angiogenesis models, and hypoxia-driven VEGF-A production.

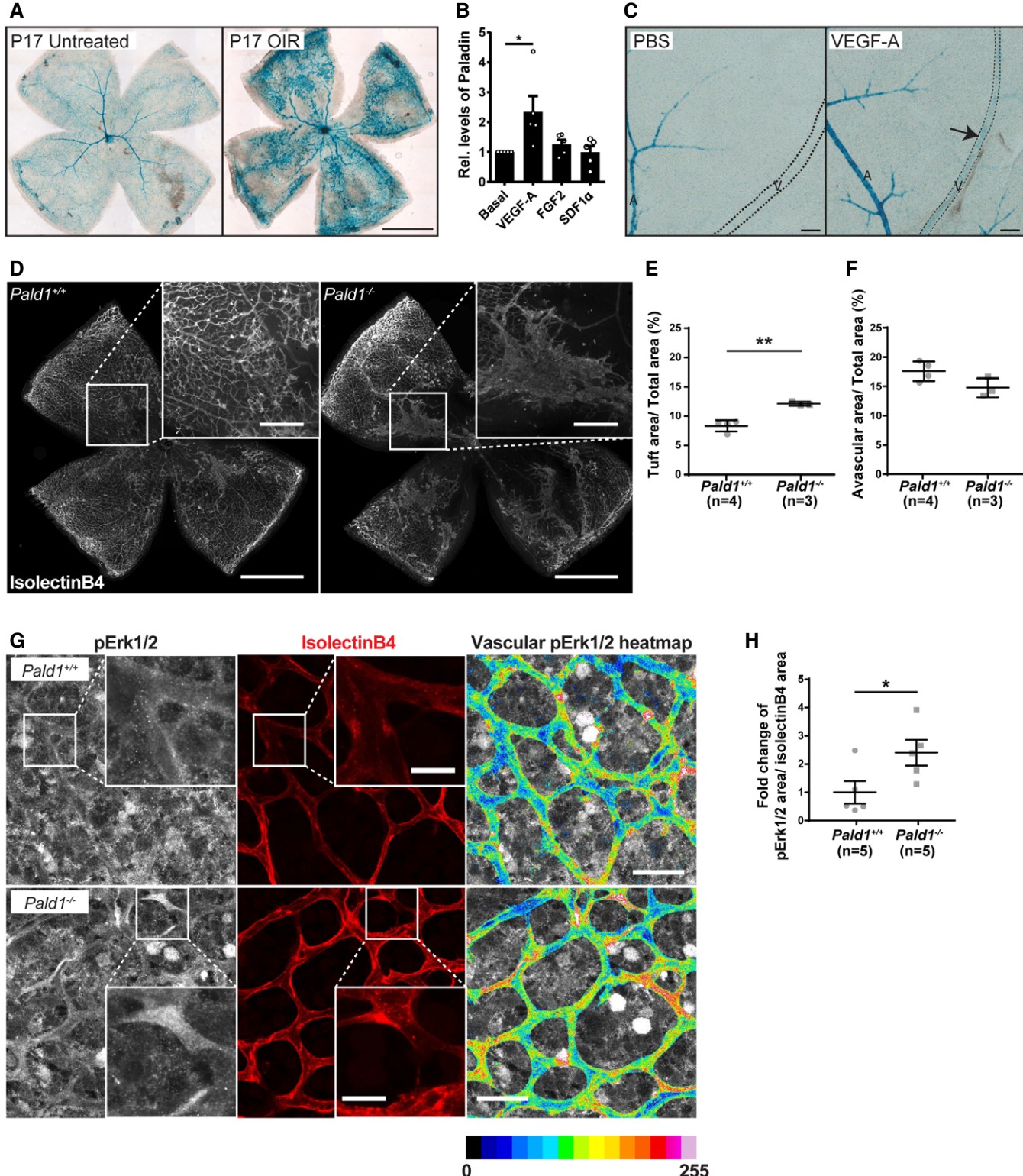

**Figure 5.**

**Figure 5. Paladin is induced by VEGF-A and regulates Erk phosphorylation in pathological angiogenesis.**

A  Eyes from $Pald1^{+/LacZ}$ mice collected at P17 from untreated animals or animals with oxygen-induced retinopathy (OIR). $Pald1$-promoter driven LacZ expression and X-gal staining generated signals in capillaries, veins, and arteries in the OIR retina at P17 compared with the normoxia control with predominantly arterial LacZ expression. Scale bar: 1 mm.

B  Paladin levels in primary human umbilical vein endothelial cells (HUVEC) untreated or treated for 24 h with VEGF-A (50 ng/ml), FGF2 (50 ng/ml), or SDF1α (30 ng/ml), quantified by immunoblotting for Paladin and β-actin (loading control). Mean ± SEM, one-way ANOVA, n = 5 biological replicates.

C  Eyes from adult $Pald1^{LacZ/+}$ mice collected at 72 h after single-bolus intravitreal injection of 1 µg VEGF-A, and PBS in the contralateral eye, followed by X-gal staining. Arrow indicates $Pald1$ promoter activity in veins (outlined by dashed line) specifically after VEGF-A treatment. A, artery; V, vein. Scale bar: 100 µm.

D–F  Representative images of isolectinB4-stained P17 retinas from OIR-challenged $Pald1^{+/+}$ and $Pald1^{-/-}$ mice (detailed view in the insets) (D). Scale bar: 1 mm (inset 250 µm). Quantification of neovascular tuft area (E) and avascular area (F). Each dot represents the mean of both retinas per mouse. Mean ± SEM, unpaired t-test, n = 3 litters, 4 $Pald1^{+/+}$, and 3 $Pald1^{-/-}$ pups.

G  Representative images of P15 retinal vasculature immunostained for isolectinB4 and pT202/pY204 Erk1/2 (pErk1/2). pErk1/2 staining within the vessels is also visualized using a 16-colour heatmap to display staining intensity. Scale bar: 30 µm (inset 10 µm).

H  Quantification of pErk1/2 immunostaining as shown in (G) within isolectinB4-positive vessels, as fold-change of pErk1/2 stained area. Each dot represents the mean of both retinas per mouse. Mean ± SEM, unpaired t-test, n = 3 litters, 5 $Pald1^{+/+}$, and 5 $Pald1^{-/-}$ pups.

Data information: *$P < 0.05$, **$P < 0.01$.
Source data are available online for this figure.

---

Potentially, Paladin might also regulate the activity of other membrane proteins and receptors. Indeed, Paladin interacts with TLR9, an entirely endosomal signalling receptor, and reduced Paladin expression leads to blunted TLR9 signalling (Li et al, 2011). Paladin has also been identified as a negative regulator of insulin receptor signalling, and PALD1 deficiency leads to increased insulin receptor levels and increased AKT downstream signalling (Huang et al, 2009). Considering the diversity of receptors affected by Paladin, its activity as a PI phosphatase may be the common denominator regulating membrane protein trafficking and thereby signalling.

In conclusion, we demonstrate that Paladin is a VEGF-A–inducible PI phosphatase that regulates endothelial sprouting and VEGFR2 trafficking and signalling, likely exerting its effects by controlling the level of PI in the early endosomal compartment and thereby affecting ERK1/2 signalling downstream of VEGF-A/VEGFR2.

# Materials and methods

## Mice

C57BL/6 mice with constitutive deletion of Pald1 (Exon 1–18 replaced by a LacZ reporter cassette) have been generated and back-crossed for at least 10 generations (Wallgard et al, 2012). $Pald1^{+/-}$ inter crosses were performed to generate homozygous and heterozygous littermates. All animal experiments were performed in compliance with the relevant laws and institutional guidelines and were approved by the Uppsala University board of animal experimentation. For in vivo signalling, VEGF-A (0.25 µg/g body weight) or peroxyvanadate (50 µmol/g body weight) was injected into the tail vein of adult (6–10 weeks old) mice followed by retrieval and lysis of hearts or lungs at specific time points and immunoblotting. MEK inhibitor U0126 (V1121, Promega) was injected intraperitoneally (5 mg/kg). For short treatment, pups were injected twice at P5 with a 2-h interval and eyes were collected 2 h after the last injection. For long treatment, pups were injected twice with a 12-h interval at P4 and retinas were collected for analyses at P5. As a vehicle, 40% DMSO in sterile 1×PBS was used.

Sample size was chosen to ensure reproducibility and allow stringent statistical analysis. Adult mice were age- and sex-matched and randomized by alternating assignment to treatment groups. Blinding of drugs for animal experiments was not performed.

## Statistical analysis

GraphPad Prism6 and Prism7 were used for statistical analysis. Normal distribution was verified through testing. Identification of statistical outliers was performed and let to the removal of one data point in Figs 4I and EV4B using ROUT method. Statistical analysis of two data sets was done by unpaired Student's t-test and of three data sets or more was done by one-way ANOVA or multiple t-test. Two-way ANOVA was used when assessing response to two factors. Sidak's or Tukey post hoc test was used to correct for multiple comparison. Sample size was chosen to ensure reproducibility and allow stringent statistical analysis. Statistical significance is indicated as follows: *$P \leq 0.05$, **$P \leq 0.01$, ***$P \leq 0.001$, ****$P \leq 0.0001$. Actual P-values are listed in Appendix Fig S2.

## Cell culture and reagents

HUVECs (ScienCell Research Laboratories) and HDMECs (Promo-Cell) were cultured in cell culture dishes coated with or without (but consistent for each experiment) 1% gelatine using endothelial cell medium MV2 (PromoCell) with all supplements (5% FCS, 5 ng/ml hEGF, 0.5 ng/ml VEGF, 20 ng/ml R3 IGF, 1 µg/µl ascorbic acid, 10 ng/ml bFGF, and 0.2 µg/µl hydrocortisone) at 37°C and 5% $CO_2$. Cells between four to seven passages were used.

Cells were treated with the following reagents: 50 ng/ml $mVEGF-A_{164}$ (PeproTech), 50 ng/ml $hVEGF-A_{165}$ (PeproTech), 50 ng/ml rh FGF2 (RD systems), 30 ng/ml rh SDF1α (Immuno-Tools), and 100 µM peroxyvanadate. HDMECs were starved for 2–6 h or overnight in 0.1% FBS prior to growth factor stimulations.

To knockdown PALD1 mRNA, semi-confluent (40–50%) HDMECs were transfected with siRNAs using 8–10 pmol/well of a 6-well plate or on an 8-well Nunc Lab-Tek II Chamber Slide, targeting PALD1 (s25894, s25895 referred to as KD1 and KD2, respectively, Ambion), or non-targeting siRNA (Stealth RNAi negative control, medium GC, Thermo Fisher) using RNAi Max (Invitrogen) according to manufacturer's instructions and cells were used for experiments 72 h later.

To determine cell proliferation rates HDMECs were seeded onto 12-well plates at 20,000 cells/well and allowed to adhere for 24 h, followed by transfection with siRNA targeting *PALD1* (s25895/KD2) or non-targeting siRNA negative control, as described above. Fresh cell culture medium was changed to the cells 24 h post-transfection. Cellular growth was thereafter monitored as a relative confluency using the IncuCyte ZOOM Live-cell Imaging System (Essen BioSciences, Ltd., Hertfordshire, United Kingdom). Images were acquired in 3-h intervals, 9 images/well, for a 72-h period using a 10× objective. Mean confluency of the cells on each well at certain time point was analysed using an overlay mask capable of measuring cellular coverage of the images. Data analysis for four replicate experiments with tripticates was done in GraphPad Prism 7.02 Software using two-way ANOVA, followed by Sidak's multiple comparison test.

### HUVEC spheroid sprouting assay

HUVECs were cultured and siRNA transfection was performed as described above. Spheroids were formed from 1,000 cells/spheroid in MV2 cell culture medium (PromoCell) containing 5% FBS and 0.4% methylcellulose (2% stock solution) using a 96-well round-bottom plate (Costar, #3788). Spheroids were collected after 24 h and resuspended in a mixture of 1.5 mg/ml type I collagen (Advanced BioMatrix, PureCol), 0.4% methylcellulose (Sigma, #M7027). 2.5% FBS in polymerization buffer [Ham's F12 medium-Glutamax (Gibco), 12.5 mM NaOH, 1.25× F12 (Gibco), 25 mM HEPES, 0.146% NaHCO$_3$. 1.25% Glutamax-I (Gibco)]. Twenty-five spheroids were seeded per well of a 24-well plate on a thin 1.5 mg/ml collagen layer. After 1 h of gel polymerization at 37°C, 50 ng/ml VEGF-A or control medium was applied. The gels were fixed after 24 h in 4% PFA for 1 h at room temperature or overnight at 4°C. Bright-field images on an inverted microscope with 5× objective were acquired, and cumulative sprout length was measured by ImageJ.

### Phosphatase assay

HEK293 cells were transfected with plasmids (pcDNA3.1 or pLenti7.3-V5 backbone) encoding V5-tagged human full-length paladin or mutant paladin (see Fig EV1A) using Lipofectamine 2000 (Invitrogen). As controls wild-type PTEN (28298 by Addgene), phosphatase-dead mutant PTEN C124S (28300 by Addgene) or β-galactosidase (pLenti7.3/V5-GW/lacZ by Invitrogen) were used. Cells were washed twice with 1×TBS and lysed [0.5% Triton X-100, 0.5% sodium deoxycholate, 150 mM NaCl, 20 mM Tris, pH 7.4, 1× protease inhibitor cocktail (Roche) or 20 mM HEPES, 150 mM NaCl, 1% NP40, 1× protease inhibitor cocktail (Roche)], and immunoprecipitation was performed with antibodies targeting the V5-tag of paladin constructs (Invitrogen) or FLAG-tag of PTEN constructs (F3165, Sigma). As a positive control in protein phosphatase assays, endogenous TC-PTP was immunoprecipitated (6F3 clone, MediMabs). After 2 h at 4°C, lysates were incubated with Protein-G sepharose beads (GE Healthcare) for 45 min at 4°C. Subsequently, beads were washed twice with lysis buffer and once with assay buffer [25 mM Tris–HCl, 140 mM NaCl, 2.7 mM KCl, 10 mM DTT, or SHIP2 reaction buffer (Echelon)] and resuspended in 100 μl (for triplicates) or 65 μl (for

duplicates) of assay buffer (one 10-cm dish of HEK293 cells per triplicate or two duplicates).

Phosphoinositide phosphates (Echelon, diC8) and inositol phosphates (Echelon, IP$_6$ by Merck) were suspended in assay buffer at 3,000 pmol/well. The protein to be tested (30 μl of immunocomplexes) was added, and the reaction was stopped after 20 min (PI (4,5)P$_2$), 30 min (PI(3,4,5)P$_3$), or after 90 min for the initial screening shown in Fig EV1C by adding an equal volume of molybdate dye solution (V2471 Promega) and after 15-min incubation at RT absorbance at 600 nm was measured. Released phosphate was calculated by comparison to the amount of free phosphate in positive control (3,000 pmol of K$_2$PO$_4$). The colorimetric assay was performed in 96-well half area plates (Costar # 3690).

Phosphopeptide phosphatase activity was assessed by the radioactive assay using src-optimal peptide and PKC-optimal peptide as previously described (Sorby *et al*, 2001).

### Immunocytochemistry

For visualising PI(4,5)P$_2$, HDMECs (20,000 cells/well) were plated on 8-well Nunc Lab-Tek II Chamber Slides and allowed to adhere to before being transfected with siRNA targeting *PALD1* or a non-targeting siRNA control. Cells were maintained for 72 h at 37°C and 5% CO$_2$ to allow for the formation of a confluent monolayer. Before beginning the experiment, cells were serum starved at 37°C in MV2 medium for 1.5 h then placed on ice for a further 1.5 h to attenuate internalization/endocytotic processes. HDMECs were then stimulated with mVEGF-A164 (50 ng/ml) for 2 or 10 min, or a PBS control at 37°C. After stimulation, cells were washed in an ice-cold PBS buffer and immediately fixed with 1% PFA in 2.5 mM triethanolamine, pH 7.5, containing 0.1% Triton X-100 and 0.1% NP-40 for 25 min at RT then permeabilized in 0.5% Triton X-100 for 10 min. Samples were then blocked with 0.2% Tween 20/3% BSA/5% FCS/0.05% Sodium Deoxycholate in PBS. Subsequently, cells were incubated with mouse anti-PI(4,5)P2 (1:200, Echelon Biosciences, Z-P045), goat anti-VE-cadherin (1:200, R&D systems, AF1002), and rabbit anti-PI (1:200, Atlas Antibodies, HPA015696) antibodies overnight at 4°C. After washing, samples were incubated with fluorophore conjugated secondary antibodies and DAPI to visualize nuclei.

To determine colocalisation for EEA1, VEGFR2, and Paladin, HDMECs (20,000 cells/well) were plated on 8-well Nunc Lab-Tek II Chamber Slides and allowed to adhere to before being transfected with siRNA targeting *PALD1* or a non-targeting siRNA control. Cells were maintained for 72 h at 37°C and 5% CO$_2$ to allow for the formation of a confluent monolayer. Before beginning the experiment, cells were serum starved at 37°C in MV2 medium 3 h before stimulation with mVEGF-A164 (50 ng/ml) for 2 or 10 min or a PBS control at 37°C. This was followed by fixation in 3% PFA for 3 min, permeabilized in 0.1% Triton X-100 for 3 min, and postfixed in 3% PFA for 15 min. Samples were then blocked with 0.2% Tween 20/3% BSA/5% FCS/0.05% Sodium Deoxycholate in PBS. For immunostaining, cells were incubated with mouse anti-EEA1 (1:200, BD Bioscience, 610457), goat anti-VEGFR2 (1:200, R&D systems, AF644), and rabbit anti-paladin (1:200, Atlas Antibodies, HPA015696) antibodies overnight at 4°C. After washing, samples were incubated with fluorophore conjugated secondary antibodies and DAPI to visualize nuclei.

Giantin staining: HDMEC were fixed with 4% PFA, permeabilized with 0.2% Triton X-100 for 10 min, and blocked in 0.2% Tween 20/3% BSA/5% FCS/0.05% Sodium Deoxycholate in PBS and incubated with anti-Giantin 1:100 (Abcam ab24586) overnight at 4°C. After washing, samples were incubated with fluorophore-conjugated secondary antibodies (Jackson Immunoresearch). Samples were mounted using Fluoromount-G (SouthernBiotech) or ProLong Gold (Invitrogen).

Images were acquired with a Leica SP8 confocal microscope, and image acquisition was done with a 63× objective. Images were randomised and blinded before being processed and quantified using ImageJ software (NIH). Images were then collated and statistics run using GraphPad.

## Proximity Ligation Assay

HDMECs (20,000 cells/well) were plated on 8-well Nunc Lab-Tek II Chamber Slide and allowed to form a confluent monolayer over 72 h at 37°C and 5% $CO_2$. The cells were serum starved at 37°C in MV2 medium 3 h before stimulation with mVEGF-A164 (50 ng/ml) for 2 or 10 min or a PBS control at 37°C. This was followed by fixation in 3% PFA for 3 min, permeabilized in 0.1% Triton X-100 for 3 min, and postfixed in 3% PFA for 15 min. Samples were blocked in Duolink blocking buffer for 2 h at 37°C and used for PLA. The Duolink protocol (Sigma-Aldrich) was followed using rabbit anti-paladin (1:200, Atlas Antibodies, HPA015696) and goat anti-VEGFR2 (1:200, R&D systems, AF644) antibodies, and oligonucleotide-linked secondary antibodies, denoted PLUS and MINUS probes. Fluorescent probes were then added which bound to the reacted oligonucleotides, indicating a proximity between VEGFR2 and Paladin. Upon completion of the PLA protocol, cells were counterstained with antibodies against VE cadherin (1:200, Santa Cruz, SC9989) to visualise cell junctions and DAPI (Thermo Fisher) to detect nuclei. As a technical control for each experiment, the same procedure was performed with the omission of either of the antibodies or either of the PLUS or MINUS probes.

Images were acquired with a Leica SP8 confocal microscope and image acquisition was done with a 63× objective. Only cells positive for VE cadherin were imaged and analysed. Images were processed and quantified with ImageJ software (NIH).

## Immunoprecipitation and Western blotting

Cells were washed once with cold 1×PBS and lysed in cell lysis buffer [0.02 M HEPES pH 7.5, 0.15 M NaCl, 1% (w/v) NP 40, 1 mM $Na_3VO_4$, in PBS, and 1× Protease Inhibitor Cocktail (Roche)].

For *in vivo* signalling study, dog VEGF-$A_{165}$ (5 µg/20 g body weight) was administrated via the tail vein and mice were sacrificed after 1–20 min circulation time, and lung and heart were removed immediately and snap frozen. Control mice received an equal volume of PBS. Snap frozen tissue was lysed in 1% NP-40, 1% sodium deoxycholate, 0.01 M NaP$_i$, 150 mM NaCl, 2 mM EDTA, 1 mM $Na_3VO_4$, 1× Protease Inhibitor Cocktail (Roche), or in 20 mM HEPES, 150 mM NaCl, 1% NP-40 with 2 mM $Na_3VO_4$ and 2.5× Protease Inhibitor Cocktail (Roche), homogenized with Tissue Tearor (BioSpec Products) and sonicated six to eight times for 5 s at 200 W (Bioruptor, diagenode). After 1-h incubation at 4°C, tissue lysates were centrifuged at 21,100 g for 20 min. Protein

concentration was measured with the BCA protein detection kit (Thermo Fisher Scientific).

For immunoprecipitation, lysates were pre-cleared for 2 h at 4°C with unspecific goat IgG (Jackson Immuno Research) and Protein-G sepharose 4 Fast Flow beads (GE Healthcare) and incubated overnight at 4°C with goat anti-mouse VEGFR2 (R&D, AF644) or goat anti-human VEGFR2 (R&D, AF357). The lysates were incubated with Protein-G sepharose beads for 1 h at 4°C and subsequently the beads were washed five times with lysis buffer and denatured in 2× sample buffer (Life Technologies) at 95°C for 5 min.

Proteins were separated on a 4–12% BisTris polyacrylamide gel (Novex by Life Technologies) and transferred to an Immobilon-P PVDF membrane (Millipore) using the Criterion Blotter system (BioRad). The membrane was blocked with 5% skimmed milk in TBS 0.1% Tween, or with 5% BSA in TBS 0.1% Tween for anti-phospho antibodies and incubated overnight at 4°C. Following antibodies were used. Rabbit anti-paladin (1:1,000; Atlas Antibodies, HPA017343), rabbit anti-phospho-VEGFR2 pY1175 (1:1,000, Cell Signaling, 2478), rabbit anti-VEGFR2 (1:1,000, Cell Signaling, 2479), rabbit anti-phospho-PLCγ pY783 (1:1,000, Invitrogen, 44-696G), rabbit anti-PLCγ (1:1,000, Cell Signaling, 2822), rabbit anti-phospho-Erk1/2 pThr202/pTyr204 (1:1,000, Cell Signaling, 4377), rabbit anti-Erk1/2 (1:1,000, Cell Signaling, 9102), rabbit anti-phospho-Akt pSer473 (1:1,000, Cell Signaling, 4060), rabbit anti-Akt (1:1,000, Cell Signaling, 9272), rabbit anti-phospho-Src pTyr416 (1:1,000, Cell Signaling, 6943), rabbit anti-phospho-Src pTyr527 (1:1,000, Cell Signaling, 2105), rabbit anti-Src (1:1,000 Cell Signaling, 2123), and goat anti-actin (1:1,000, Santa Cruz, sc1615). Membranes were washed in TBS 0.1% Tween and incubated with horseradish peroxidase (HRP) conjugated secondary anti-rabbit (1:10,000, GE Healthcare) or anti-goat antibodies (1:10,000, Invitrogen), respectively. Membranes were washed in TBS 0.1% Tween and developed using ECL prime (GE Healthcare). Luminescence signal was detected by the ChemiDoc MP system (Bio-Rad) and densitometry performed using Image Lab software (Bio-Rad).

## Surface biotinylation assay

For assessment of surface-bound VEGFR2 levels after VEGF-A stimulation, siRNA transfected HDMECs were starved for 2 h in basic endothelial cell medium (PromoCell) with only 0.1% FBS and stimulated with recombinant VEGF-$A_{164}$ (50 ng/ml) for indicated time points. Cells were washed twice with cold 1×DPBS (containing $Mg^{2+}$ and $Ca^{2+}$) and biotinylated with 0.5 mg/ml EZ-Link Sulfo-NHS-Biotin (Thermo Scientific) in DPBS at 4°C for 45 min with gentle shaking. The reaction was stopped by washing twice with cold DPBS and incubation with cold 100 mM glycine in DPBS for 10 min on ice. Subsequently, the cells were washed and lysed in modified RIPA buffer (20 mM HEPES, 150 mM NaCl, 1% NP-40) with protease (Roche) and phosphatase inhibitors (1 mM $Na_3VO_4$).

For assessment of the VEGFR2-internalized pool after VEGF-A stimulation, biotinylation of cells surface receptors was performed prior to VEGF-A stimulation as described above. Cells were stimulated with VEGF-$A_{164}$ (50 ng/ml) for indicated time points. Cells were washed in cold 1× DPBS and cell surface biotin was cleaved off by incubating the cells on ice with 100 mM of membrane impermeable reducing agent MESNA (2-mercaptoethane sulphonic acid) (Sigma) in stripping buffer [50 mM Tris, pH 8.6, 150 mM NaCl,

1 mM EDTA, 0.2% BSA (pH 8.6)] for 3× 15 min. After washing, cells were lysed with RIPA buffer as described above.

Equal amounts of protein lysates were immunoprecipitated with streptavidin sepharose beads (GE Healthcare) overnight at 4°C after which beads were washed extensively with RIPA buffer and suspended in 2× NuPAGE LDS Sample Buffer (Invitrogen) with NuPAGE Sample Reducing Agent (Invitrogen). Protein separation and Western blotting were performed as described above.

### Retina preparation, whole mount staining, and imaging

Eyes were harvested and either fixed in 4% PFA for 10 min at RT, dissected and postfixed in ice-cold methanol for at least 2 h (pErk; for filopodia analysis), or fixed with 2% PFA for 5 h (Erg) or 4% PFA for 1 h at RT (Cyclin D1). After rehydration retinas were permeabilized and blocked (0.1–0.5% Triton X-100, 0.05% sodium deoxycholate, 1% BSA, 2% FBS, 0.02% sodium azide in PBS, or 0.3% Triton X-100, 3% FBS, 3% donkey serum) for 1–2 h at RT and stained overnight at 4°C using the following antibodies: rabbit anti-Erg (1:300, Abcam, ab92513), rabbit anti-cyclinD1 (1:50, Thermo Scientific RM-9104), rabbit anti-pERK1/2 (1:100, Cell Signaling #9101), rat anti-VEC (1:100, BD 555298), and rabbit anti-pY685 VEC [1:50, kind gift from Elisabetta Dejana (Orsenigo *et al*, 2012)]. After washing, retinas were incubated with appropriate fluorophore-coupled secondary antibodies and fluorophore conjugated isolectinB4 (Jackson Immunoresearch) or washed in PBlec (1% Triton X-100, 0.1 mM $CaCl_2$, 0.1 mM $MgCl_2$, 0.1 mM $MnCl_2$ in PBS at pH 6.8) for at least 1 h at RT and stained with biotinylated isolectinB4 (Sigma) overnight at 4°C and incubated with Streptavidin-Alexa 488 (Invitrogen). Retinas were flat-mounted in Fluoromount-G (Southern Biotech), or in ProLong Gold (Invitrogen).

Images were acquired with LSM700, AxioImager M2 microscope (Zeiss), or Leica SP8 confocal microscopes. Image acquisition was done with 5× (for the tile scans), 10×, 20×, 40×, and 63× objectives. Images were processed and blinded before quantification using ZEN software (Zeiss), LAS (Leica), ImageJ software (NIH), or Cell Profiler (Broad Institute), USA (Lamprecht *et al*, 2007).

### Oxygen-induced retinopathy model

A litter of P7 pups and their mother and/or foster mother were exposed to 75% oxygen in a semi-sealed oxygen chamber (ProOx 110 sensor and A-Chamber, Biospherix, Parish, NY) from P7 to P12, followed by room oxygen from P12 to P15–P17. Pups were sacrificed at indicated time point, and the eyes were collected. To assess vascular permeability following OIR, P17 pups were warmed under a heat lamp and then 50 μl of green fluorescent microspheres (1% solution of 25 nm FITC-conjugated microsphere Fluoro-MAX G25, Thermo Scientific, Fremont, CA) were injected via tail vein using a 30-gauge insulin syringe while the mice were under temporary isoflurane anaesthesia. Following injection, microspheres were allowed to circulate for 15 min before the mice were once more placed under isoflurane anaesthesia. PBS was flushed through the vasculature via cardiac perfusion to remove excess microspheres, which had not extravasated followed by 4% PFA for tissue fixation. Eyes were enucleated and fixed in 4% PFA at RT for 30 min before the retinas were dissected and immunostained with isolectinB4 conjugated to Alexa647 and pERK1/2 as described.

To visualize *Pald1* expression after OIR, eyes from P17 pups were collected and processed for lacZ staining as described below.

### Intravitreal injections and lacZ staining of retinas

Ten-week-old female mice were anesthetized with isofluorane (AbbVie, Sweden). Prior to injection, the pupil was dilated with a drop of tropicamide (0.5% mydriacyl). A single injection of 1 μg dog VEGF-$A_{164}$ (1 μl injection volume) into the intravitreal space was done using a Hamilton syringe (Microliter #701RN with 34 gauge/25 mm/pst4 removable needle). The eyes were collected 72 h after injection and processed for lacZ staining.

For lacZ staining eyes were fixed in 0.4% PFA for 4 h at RT, retinas were dissected and permeabilized by washing three times for 20 min with detergent rinse (2 mM $MgCl_2$, 0.01% sodium deoxycholate, 0.02% Nonidet P-40, PBS). Retinas were stained with 1 mg/ml x-gal (Promega) diluted in staining solution (detergent rinse containing 5 mM potassium ferricyanide, 5 mM potassium ferrocyanide) at 37°C overnight, protected from light. Retinas were washed twice for 10 min in detergent rinse, followed by two PBS washes and post-fixation with 4% PFA for 1 h at RT and mounted in Fluoromount-G (Southern Biotech). Images were acquired with AxioImager M2 (Zeiss).

### Quantitative real-time PCR

Eyes were collected from P4 and P5 pups and placed in RNA*later*® solution (Ambion) immediately after collection. RNA was isolated using the RNeasy Micro Kit (Qiagen) and processed for quantitative real-time PCR as described previously (Wallgard *et al*, 2012). The TaqMan Assays (Applied Biosystems) used *Gapdh* (4352932E) and *Ccnd1* (Mm00432359_m1, $n = 10/11$).

## Data availability

This study includes no data deposited in external repositories.

**Expanded View** for this article is available online.

### Acknowledgements

We thank Kurt Ballmer-Hofer for kindly sharing canine VEGF-$A_{164}$. We thank Elisabetta Dejana for the kind gift of pY685 VE-cadherin antibodies. We thank Carina Hellberg (deceased), Katie Bentley, and Andrew Philippides for valuable discussion and input. Bright field and fluorescent images were acquired at the BioVis imaging facility at The Rudbeck Laboratory, Uppsala University, Sweden. This work was supported by the Swedish Cancer Foundation, Beijer Foundation, Åke Wiberg's Foundation, Magnus Bergwall's Foundation, and Swedish Research Council (E.E., L.C.-W.); Knut and Alice Wallenberg Foundation (L.C.-W.); and Gustav Adolf Johansson Foundation (D.T.L., J.L., I.E.).

### Author contributions

Experimental design, generation, and analysis of data: AN, RP, DTL, CT, TN, ROS, EE, JL, FPR, IE, SJ, LC-W, MH; Manuscript figure assembly: AN, RP, DTL, CT, ROS; Reagents: PB; Supervision: LC-W, MH; Manuscript writing: AN, RP, CT, LC-W, MH.

 

## Conflict of interest

The authors declare that they have no conflict of interest.

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
