## [Review Process File · EMBO Reports]

Paladin is a phosphoinositide phosphatase regulating endosomal VEGFR2 signalling and angiogenesis

Anja Nitzsche, Riikka Pietila, Dominic T Love, Chiara Testini, Takeshi Ninchoji, Ross O Smith, Elisabet Ekvarn, Jimmy Larsson, Francis P Roche, Isabel Egana, Suvi Jauhiainen, Philipp Berger, Lena Claesson-Welsh, and Mats Hellström

DOI: [10.15252/embr.202050218](https://doi.org/10.15252/embr.202050218)

Corresponding author(s): Mats Hellström (mats.hellstrom@igp.uu.se)

Review Timeline:

Submission Date:	13th Feb 20
Editorial Decision:	5th Mar 20
Revision Received:	1st Oct 20
Editorial Decision:	30th Oct 20
Revision Received:	7th Nov 20
Accepted:	18th Nov 20

Editor: Deniz Senyilmaz Tiebe

Transaction Report:

Dear Mats,

Thank you for the submission of your research manuscript to our journal, which was now seen by two referees, whose reports are copied below.

As you can see, the referees express interest in the proposed role of Paladin in regulation of angiogenesis as a phosphoinositide phosphatase. However, they also raise a number of concerns that need to be addressed to consider publication here. In particular, the referees point out

- That deeper analysis on the role of Paladin in regulation of VEGFR2 recycling/trafficking is required (ref #1 paragraphs 6, 7 and ref #2 point 1)
- That better characterization of the role of Paladin in VEGFA signalling is necessary (ref #1 paragraph 8, ref #2 point 2).
- To some discrepancies in the data, missing controls and quantifications.

I find the reports informed and constructive, and believe that addressing the concerns raised will significantly strengthen the manuscript.

Given these constructive comments, we would like to invite you to revise your manuscript with the understanding that the referee concerns (as in their reports) must be fully addressed and their suggestions taken on board. Please address all referee concerns in a complete point-by-point response. Acceptance of the manuscript will depend on a positive outcome of a second round of review. It is EMBO reports policy to allow a single round of revision only and acceptance or rejection of the manuscript will therefore depend on the completeness of your responses included in the next, final version of the manuscript.

1. A data availability section providing access to data deposited in public databases (or stating that no data was deposited) is missing.
2. Your manuscript contains statistics and error bars based on $n=2$ or on technical replicates. Please use scatter plots in these cases.

Supplementary/additional data: The Expanded View format, which will be displayed in the main HTML of the paper in a collapsible format, has replaced the Supplementary information. You can submit up to 5 images as Expanded View. Please follow the nomenclature Figure EV1, Figure EV2 etc. The figure legend for these should be included in the main manuscript document file in a section called Expanded View Figure Legends after the main Figure Legends section. Additional Supplementary material should be supplied as a single pdf labeled Appendix. The Appendix includes a table of content on the first page with page numbers, all figures and their legends. Please follow the nomenclature Appendix Figure Sx throughout the text and also label the figures according to

this nomenclature. For more details please refer to our guide to authors.

2) individual production quality figure files as .eps, .tif, .jpg (one file per figure).

3) a .docx formatted letter INCLUDING the reviewers' reports and your detailed point-by-point responses to their comments. As part of the EMBO Press transparent editorial process, the point-by-point response is part of the Review Process File (RPF), which will be published alongside your paper. For more details on our Transparent Editorial Process, please visit our website: <https://www.embopress.org/page/journal/14693178/authorguide#transparentprocess> You are able to opt out of this by letting the editorial office know (emboreports@embo.org). If you do opt out, the Review Process File link will point to the following statement: "No Review Process File is available with this article, as the authors have chosen not to make the review process public in this case."

4) a complete author checklist, which you can download from our author guidelines (). Please insert information in the checklist that is also reflected in the manuscript. The completed author checklist will also be part of the RPF.

5) Please note that all corresponding authors are required to supply an ORCID ID for their name upon submission of a revised manuscript (). Please find instructions on how to link your ORCID ID to your account in our manuscript tracking system in our Author guidelines ().

6) We replaced Supplementary Information with Expanded View (EV) Figures and Tables that are collapsible/expandable online. A maximum of 5 EV Figures can be typeset. EV Figures should be cited as 'Figure EV1, Figure EV2' etc... in the text and their respective legends should be included in the main text after the legends of regular figures.

7) We would also encourage you to include the source data for figure panels that show essential data.

Numerical data should be provided as individual .xls or .csv files (including a tab describing the data). For blots or microscopy, uncropped images should be submitted (using a zip archive if multiple images need to be supplied for one panel). Additional information on source data and instruction on how to label the files are available .

8) Our journal encourages inclusion of *data citations in the reference list* to directly cite datasets that were re-used and obtained from public databases. Data citations in the article text are distinct from normal bibliographical citations and should directly link to the database records from which the data can be accessed. In the main text, data citations are formatted as follows: "Data ref: Smith et al, 2001" or "Data ref: NCBI Sequence Read Archive PRJNA342805, 2017". In the Reference list, data citations must be labeled with "[DATASET]". A data reference must provide the database name, accession number/identifiers and a resolvable link to the landing page from which the data can be accessed at the end of the reference. Further instructions are available at .

9) Please make sure to include a Data Availability Section before submitting your revision - if it is not applicable, make a statement that no data were deposited in a public database. Primary datasets (and computer code, where appropriate) produced in this study need to be deposited in an appropriate public database (see).

The accession numbers and database should be listed in a formal "Data Availability " section (placed after Materials & Method) that follows the model below. Please note that the Data Availability Section is restricted to new primary data that are part of this study.

Data availability

10) Regarding data quantification, please ensure to specify the name of the statistical test used to generate error bars and P values, the number (n) of independent experiments underlying each data point (not replicate measures of one sample), and the test used to calculate p-values in each figure legend. Discussion of statistical methodology can be reported in the materials and methods section, but figure legends should contain a basic description of n, P and the test applied.

Please note that error bars and statistical comparisons may only be applied to data obtained from at least three independent biological replicates.

I look forward to seeing a revised version of your manuscript when it is ready. Please let me know if you have questions or comments regarding the revision.

Yours sincerely,

Deniz

Deniz Senyilmaz Tiebe, PhD
Editor
EMBO Reports

Referee #1:

In this paper the authors provide evidences that paladin dephosphorylates PIPs and this activity contributes both in vitro and in vivo to regulate VEGFR2 activity through affecting its trafficking. The data presented are interesting but some of them require more controls and experiments to better define their meaning.

Fig 1. In this figure the authors show that paladin dephosphorylates both PIP 2 and PIP3 . Because PIP3 is mainly active at plasmamembrane and paladin is present in endosomal compartment, I think that it is quite surprising that paladin work s on PIP3. This point has to be explained. I suggest to define the Km of these substrates . I think this data is vey important to identify the relevant physiologic substrate of paladin.

Furthermore it is necessary to add PI 5 phosphatase as a second positive control specific for PIP2.

The absence of colocalization between paladin and VE-cadherin is well documented in S1. However PIP2 may accumulate in caveolae (e.g. 10.1073/pnas.0900216106; 10.1074/jbc.M110.196022) and it is well established that VEGFR2 is present in this kind of plasma membrane domain type (e.g. Mol Biol Cell. 14, 334, 2003) . So I suggest to verify this possible localization of paladin in plasmamebrane .

Fig 2A. May the authors show the Co-IP experiment without overexpressing paladin? I'm aware that sometimes is very difficult to show a protein-protein interaction in native conditions. However it's the most convincing experiment to support a relevant biological interaction. Perhaps another technique may help to confirm the interaction between paladin and VEGFR2 shown panel B (Fret, in situ PLA)

Fig 2D. These data have to be quantified. Furthermore the authors have to determine where this interaction occurs (Rab4, -7, -11 positive vesicles, Golgi membrane)

Fig 2g,h. When cells are stimulated by VEGF, VEGFR2 undergoes phosphorylation. Does paladin silencing modify the internalization of phosphorylated form of VEGFR2?

Does paladin silencing affect VEGFR2 recycling to the plasma-membrane? In my opinion this issue has to be faced to offer to the readership a more compelling vision of the effect of paladin on VEGFR2 trafficking

The experiments shown in Figure 3 are fine but they need a biological counterpart, which is also useful to interpret the in vivo data. Do silencing and over-expression of paladin modify the mitogen and motogen activity of VEGFA?

Fig 3e. The in vivo analysis clearly demonstrates that paladin deletion in heart EC accelerates VEGFR2 degradation after VEGF stimulation. This effect does not occur in vitro (3A). Why? May it depend on different time courses analyzed? Which is the mechanism sustain the faster degradation observed in vivo in Pdl null mice?

Fig 4 & 5. Does Erk inhibition rescue the effect in retina vascularization observed in Pdl null mice? There a lot of compound in vivo tested that could easily exploited (e.g. refametinib, GDC-0994)

Referee #2:

The manuscript demonstrates that Paladin can bind to phosphoinositides (PIs) and acts as a phosphatase for PI(4,5)P₂ and PI(3,4,5)P₃. Authors also revealed that Paladin fine-tunes VEGFR2 intracellular trafficking in endothelial cells. In addition, Paladin was found to negatively regulate activation of VEGFR2 and its downstream target ERK1/2 in endothelial cells in vitro and in vivo. In accordance, Paladin germ-line KO animals showed defects in both developmental and pathological angiogenesis in the mouse retina. These observations are novel and describe new roles for Paladin. Moreover, authors highlighted the relevance of Paladin for VEGFR2 signaling. However, there are several major and minor issues that need to be addressed prior to publication:

Major issues:

1. The authors show that Paladin is a phosphatase for PIs and that Paladin affects VEGFR2 trafficking. However, importantly, the authors fail to demonstrate that the effect of Paladin on VEGFR2 trafficking is through its phosphatase activity on PIs. To test this, the authors should perform experiments confirming that Paladin regulates VEGFR2 signalling and trafficking via its activity on PIs.
2. Figure 2 A, B, C and lines 161-163 of the main text: "We observed the formation of a Paladin-VEGFR2 complex in response to VEGF-A treatment both in vitro (in primary endothelial cells) and in vivo (in a mouse model)."
This claim is not supported by data. Both in vivo and in vitro data show that VEGF alone is not sufficient to induce complex formation. This only occurs in case of peroxivanadate treatment. Moreover, peroxivanadate alone was sufficient to induce complex formation in vivo. This condition, is a important missing control in Figure 2A and 2B. Input samples should also be included. Along the same line of thought, based on Figure 2D, authors claimed that "Accordingly, super-resolution microscopy analysis confirmed VEGF-A induced co-localization of Paladin and VEGFR2. This indicated that Paladin could be involved in VEGF-A/VEGFR2 signaling." (lines 165-167). Yet, authors do not show conditions with or without VEGF stimulation, and no quantification is provided to support the claim on the levels of co-localization.
3. Figure 2H shows that Paladin KD cells have higher levels of internalised VEGFR2 upon 5-15min of VEGFA stimulation than control cells. However, quantification in Figure 2J and Sup.Figure2F shows that both the number of VEGFR2 vesicles and the number of VEGFR2 vesicles colocalising with EEA1 at 10min is equivalent to those in Paladin KD cells, whilst one would expect to have higher levels. Can authors comment on these contradictory results? Can authors provide additional confirmatory experiments to clarify this important claim in the manuscript? Moreover, the authors should complement their analysis for VEGFR2 trafficking with additional vesicular markers (Rab5+, Rab7+, Rab4+ or Rab11+ vesicles) in a time course manner to assess the fate of VEGFR2-positive

vesicles in control and Paladin KD cells.

4. In Figure 3, authors show that Paladin KD cells have higher signal transduction in response to VEGFA stimulation *in vitro* and *in vivo*. However, data presentation is very puzzling. For instance, Figure3i,j has different merged time points than Figure3f-h. Authors should provide a consistent way to display time in their analyses. The reviewer suggests that every time point should be displayed individually. In addition, quantification for pVEGFR2 levels *in vivo* is missing.

5. Figure 4: To understand the extent of the observed rescue upon use of U0126 in Paladin KO animals, it would be important to provide representative images of those that were used for the quantifications represented in these graphs. Also, in figure 4I it is not clear in which condition CyclinD1 stainings were performed (WT or KO?). Representative images for both WT and KO animals should be shown. Moreover, authors could use ERG staining to label endothelial nuclei and thus increase the precision of the quantification (CyclinD1/ERG Double-positive cells), thus excluding confounding contribution of pericytes and microglia.

6. In general, it would be more informative to show quantification of vascular parameters in Paladin WT and KO retinas in absolute numbers and not as relative to control. Moreover, authors should show all data points (such as in Figure 1B/C), avoiding bars, and privileging scatter dot plots or box and whiskers.

7. Discussion (line 297-299): "Our data suggest that Paladin is a part of a VEGF-driven negative feedback loop in retinal angiogenesis where VEGF-A upregulates Paladin which acts to dampen VEGFR2 driven signaling and endothelial sprouting"

The authors show that VEGF increases Paladin expression in HUVECS and in the mouse retina upon tail vein injection of VEGF. They do not show the existence of a loop that is interrupted if Paladin is knocked down/ knocked out. Therefore, we believe that in the discussion, the authors should rephrase their sentence to avoid overstating the significance of their findings.

Minor issues:

1. Figure S1E: It was shown that recombinant Paladin binds specifically to PI(3)P, PI(4)P, and PI(5)P, and PI(3,5)P2. It is somehow surprising not to find Paladin targets [PI(4,5)P2 and PI(3,4,5)P3] as binders. Could the authors explain these results?

2. Line 172-173: "However, the receptor was degraded at the same rate as control-treated cells after VEGF-A stimulation (Figure 2e, Suppl Figure 2b)." It is unclear how the authors concluded about degradation rates. Could the authors clarify?

3. It would be interesting to provide images of P5 retinas images with higher magnification of tip cells from lacZ-stained Paladin het animals, similar to Figure 5a/c.

4. Figure 4: To strengthen their findings, it would be relevant to show the expression pattern of VEGFR2 and pVEGFR2 in the wt and Pald1 KO retinas.

5. Line 144-145: "Rab4, -7, and -11, markers of fast recycling, slow recycling and late endosomes, respectively". This statement is incorrect. Rab7 marks late endosomes and while Rab11 marks slow recycling endosomes. The sentence should be corrected.

6. Line 147-148: "Super-resolution microscopy revealed that one-quarter of the Rab4- or Rab11-positive structures were also positive for Paladin". The quantification for the number of vesicles that

are double positive for Paladin/Rab4 and for Paladin/Rab11 should be showed in a graph.

7. Line 170-171: "siRNA-mediated knockdown of PALD1 in HDMEC resulted in a 35-51% increase of the total basal VEGFR2 pool". It is unclear what technique was used to obtain this result. It is not stated in the main text nor in the figure legend. If the technique used was Western Blot, it would be important to show the image with the bands that allowed to make this quantification.

8. Line 237-238: For clarity, the authors should clarify in the main text which animal model was used. From the main text it seems that an inducible endothelial specific knock out animal was used. But from reading the materials and methods, that does not seem to have been the case.

9. Line 342: this reference "Lanahan, 2010" has not been included in the bibliography section.

10. Line 346: "We also observed increased pTyr1173 phosphorylation in HDMEC". For clarity, the authors should rephrase this sentence to be more accurate, namely increased in which condition as compared with what.

11. Figure 1D, right panel: the expression pattern of paladin and PI4P seem to be heterogeneous across the cell population and perhaps inversely correlated. Is that the case? Could the authors show separate panels for each colour? Could the authors comment on this?

12. Figure 4K and L: Could the authors clarify why two regimes of U0126 administration were used?

13. Line 271: "VEGF-A induced production of the Paladin protein in endothelial cells in vitro and in the retinal vasculature in vivo, as indicated by LacZ reporter expression (Figure 5b,c and Suppl Figure 5a)." Authors should clarify this text as it suggests that quantifications were performed on LacZ reporter, yet, Figure 5B seems to be WB from bands showed in Sup Figure 5a.

14. Figure S1D: it is not very clear what portion of the cell this image is reporting. The authors should provide an additional image with the zoom out of this cell with the location of the zoom in marked.

Point-by-point letter MS EMBOR-2020-50218V1

Dear Dr Senyilmaz,

Thank you for your email of March 5 2020, with the reviewers' report on our manuscript by Nitzsche et al. We sincerely appreciate the constructive comments from yourself and the reviewers. We have performed an extensive revision resulting in a very substantial consolidation of the finding that Paladin is a phosphoinositide phosphatase which targets PI(4,5)P₂, with important consequences for endothelial cell biology which we investigate using *in vitro* and *in vivo* models. In the revision, we have focused on the impact of Paladin on the early events of VEGFR2 trafficking after VEGF-A stimulation. The major findings supporting an important role for Paladin in the early steps of VEGFR2 internalization are the following:

- Rapid VEGF-A induced Paladin and VEGFR2 colocalization in the cell periphery but not at junctions/membrane.
- VEGF-A-induced co-localization between Paladin and the early endosome marker EEA1.
- Augmented VEGFR2 internalization in response to VEGF-A in Paladin-deficient conditions, accompanied by increased levels of pVEGFR2 and pErk1/2.
- Marked accumulation of PI(4,5)P₂ already at 2 min of VEGF-A stimulation in Paladin-knockdown cells, supporting an important role for Paladin in PI(4,5)P₂ dephosphorylation.

In the initial submission of our work, there were some concerns relating to different effect of Paladin-deficiency when comparing *in vitro* and *in vivo* models. Overall, the *in vitro* and *in vivo* data from Paladin loss of function models are consistent with a few exceptions: 1) elevated baseline VEGFR2 *in vitro* under Paladin-deficiency but unchanged *in vivo*, 2) delayed VEGFR2 degradation *in vivo* when compared to *in vitro* and 3) delayed pErk1/2 increase *in vivo* compared to *in vitro*. Given the different contexts and signaling kinetics of the models, it is in our view still compelling that lack of Paladin consistently promoted VEGF-A production, Erk1/2 activation and angiogenesis across models *in vitro* and *in vivo* and of both physiological and pathological angiogenesis. This is also discussed in the manuscript on page 15.

Summary of changes to Main Figures:

Figure 1: previous 1b is now 1a and 1b-h are new.

Figure 2: previous 2e-h are now 2a-d and 1e-h are new.

Figure 3: 3a-e remain, previous 3f and g have been merged to 3f and one new graph added to 3f.

Figure 4: 4a-h remain, previous 4j is now 4i. Previous 4k,l are now 4l,m, Previous 4i has been modified and moved to EV 4. Figure 4j,k are new.

Figure 5: No changes.

Please find a point-by-point response to the reviewer's questions.

Referee #1:

In this paper the authors provide evidences that paladin dephosphorylates PIPs and this activity contributes both *in vitro* and *in vivo* to regulate VEGFR2 activity through affecting its trafficking. The data presented are interesting but some of them require more controls and experiments to better define their meaning.

Fig 1. In this figure the authors show that paladin dephosphorylates both PIP 2 and PIP3 . Because PIP3 is mainly active at plasmamembrane and paladin is present in endosomal compartment, I think that it is quite surprising that paladin works on PIP3. This point has to be explained. I suggest to define the Km of these substrates . I think this data is very important to identify the relevant physiologic substrate of paladin. Furthermore it is necessary to add PI 5 phosphatase as a second positive control specific for PIP2.

The *in vitro* substrate specificity of phosphoinositide phosphatase is often not absolutely strict due to structural constraints. For example, myotubularins dephosphorylate PI3P and PI(3,5)P₂ (Berger et al., Hum Mol Genet. 2002 PMID: 12045210). The *in vivo* specificity also depends on the local context availability of the substrate. We saw efficient dephosphorylation (using Paladin wt compared to C/S) mainly for PI(4,5)P₂. Moreover, we also noted colocalization of Paladin with early endosomes. From this we conclude that PI(4,5)P₂, which is transported from the plasma membrane to early endosomes, is probably the main *in vivo* substrate of paladin. In line with this, the PI(4,5)P₂ signal increased in intact *PALD1* knock-down cells treated with VEGF-A, compared to control cells (new Figure 2g,h). Although interesting and important for further understanding of the biochemical properties of Paladin's phosphatase activity, we consider it beside the scope of our study and also not within our expertise, to do *in vitro* measurements of Paladin's catalytic activity.

As the reviewer rightly points out, PI(4,5)P₂ (and PIP3) is considered to be mainly active at the plasma membrane. However, it is also important to note that those phosphoinositides are also present at other location in the endosome compartment, albeit at lower levels as also discussed in the manuscript.

The suggestion to use additional positive controls for PIP2 is good, however, our positive control PTEN shows that the assay *per se* works and also generates a value we can benchmark to. Instead our efforts in the revision have been on defining the role of Paladin in VEGFR2 turnover and signaling in endothelial cells.

The absence of colocalization between paladin and VE-cadherin is well documented in S1. However PIP2 may accumulate in caveolae (e.g. 10.1073/pnas.0900216106; 10.1074/jbc.M110.196022) and it is well established that VEGFR2 is present in this kind of plasma membrane domain type (e.g. Mol Biol Cell. 14, 334, 2003) . So I suggest to verify this possible localization of paladin in plasmamebrane .

This is a very good suggestion and we stained HDMEC for Paladin and Caveolin 1 +/- VEGF-A but failed to detect co-localization. Moreover, in-depth analyses shown in Figures 1b,c,g convince us that Paladin is localized in intracellular vesicles, not at the plasma membrane.

Fig 2A. May the authors show the Co-IP experiment without overexpressing paladin? I'm aware that sometimes is very difficult to show a protein-protein interaction in native conditions. However, it's the most convincing experiment to support a relevant biological

interaction. Perhaps another technique may help to confirm the interaction between paladin and VEGFR2 shown panel B (Fret, in situ PLA)

We thank the reviewer for this suggestion. We have now performed in situ proximity ligation assays (PLA) to confirm a VEGF-A-induced interaction between VEGFR2 and Paladin, see Figure 1e,f, and Figure EV1i. VEGF-A-induced complex formation is significantly induced at 2 min, but less so at 10 min of treatment. Blotting for Paladin on VEGFR2 IPs *in vivo* and *in vitro* confirmed complex formation but only when cells were treated with VEGF-A and peroxyvanadate (see Figure EV1).

Fig 2D. These data have to be quantified. Furthermore the authors have to determine where this interaction occurs (Rab4, -7, -11 positive vesicles, Golgi membrane)

The PLA experiments (Figure 1 e,f) revealed an interaction between VEGFR2 and Paladin that occurs quickly (at 2 min) upon VEGF-A stimulation. Based on this we performed new and improved staining and image analysis for Paladin and EEA1 and can now show VEGF-A-induced increase in Paladin/EEA1 co-localization (Figure 1g,h). Further studies then focused on the early interaction with VEGFR2, and the localization and function of Paladin (Figure 2). The localization of Paladin in other endosomal compartments is still relevant, but not the focus of the current study and thus we have removed these panels.

Fig 2g,h. When cells are stimulated by VEGF, VEGFR2 undergoes phosphorylation. Does paladin silencing modify the internalization of phosphorylated form of VEGFR2?

Yes, pVEGFR2 follows the same trend as the total VEGFR2 with more pVEGFR2 internalized after stimulation, please see included Figure (Y-axis is relative internalization and X-axis time in min, n=2). As pVEGFR2 and total VEGFR2 followed the same pattern and the internalized pVEGFR2 levels are low and therefore difficult to detect and quantify, we decided to focus on the total VEGFR2 levels.

Does paladin silencing affect VEGFR2 recycling to the plasma-membrane? In my opinion this issue has to be faced to offer to the readership a more compelling vision of the effect of paladin on VEGFR2 trafficking

We chose to focus on the effect of loss of function of Paladin at early time points of VEGFR2 trafficking. We observe an interaction of Paladin and VEGFR2 at 2 min after VEGF-A stimulation. In the absence of *PALD1*, we detect a rapid increased internalization of the receptor, increased levels of pVEGFR2 and increased co-localization of EEA1/VEGFR2 as well as an increase in PIP2 levels. In addition, the surface levels of VEGFR2 did not differ between control and *PALD1* knock-down cells even at late time points, as we followed VEGFR2 levels to 180 min after VEGF-A stimulation. Our conclusion is that Paladin regulates early internalization and trafficking of VEGFR2. We have therefore focused on the early internalization events regulated by Paladin, in this study. However, it does not rule out that Paladin have other effects on subsequent trafficking of VEGFR2 including recycling, which we also discuss, see page 14-15

The experiments shown in Figure 3 are fine but they need a biological counterpart, which is also useful to interpret the *in vivo* data. Do silencing and over-expression of paladin modify the mitogen and motogen activity of VEGFA?

We have now included data from *in vitro* studies supporting a role for Paladin in endothelial sprouting and proliferation. VEGF-A induced endothelial sprouting was enhanced in the absence of *PALD1* as compared to control siRNA, see Figure EV 4b. Endothelial proliferation was increased in *PALD1* knock-down cells compared to controls cells when tracked for 72h using Incucyte, Figure EV 4c.

Fig 3e. The *in vivo* analysis clearly demonstrates that paladin deletion in heart EC accelerates VEGFR2 degradation after VEGF stimulation. This effect does not occur *in vitro* (3A). Why? May it depend on different time courses analyzed? Which is the mechanism sustain the faster degradation observed *in vivo* in *Pdl* null mice?

The *in vivo* data from heart shows a transient delay in degradation of VEGFR2 in the *Pald1*^{-/-} mice. The kinetics of signaling and receptor degradation are different between cell- and animal experiments, with faster kinetics *in vivo* compared to cells which is as expected for example due to the higher and more consistent *in vivo* temperature. On the other hand, handling of the animals which we strive to conduct in a consistent manner, may induce stress hormones that may influence the analyses. Based on our experience from VEGFR2 and many other signaling systems, we do not expect a 100% concordance between cell and animal experiments. However, the *in vitro* analyses are important as they allow to address mechanistic aspects. In the paper we describe a consistent signaling alteration in *PALD1* silenced cells *in vitro*, and in the *Pald1* knock-out *in vivo*, in the developing retina, in the heart and in the pathological angiogenesis of the eye, allowing the conclusion that Paladin regulates early steps of internalization of VEGFR2 with consequence for signaling preferentially in the Erk1/2 pathway. The faster internalization rate seen in Paladin-deficiency agrees with the notion that VEGFR2 needs to escape from cell surface localized protein tyrosine phosphatases in order to preserve phosphorylation on key tyrosine residues such as Y1173, which is a prerequisite for downstream signaling in the Erk1/2 pathway. The discrepancy between kinetics of degradation *in vivo* and *in vitro* are likely to be due to differences in internalization kinetics between HDMEC vs heart EC or *in vitro* vs. *in vivo*). We believe that further addressing such kinetics is beyond the scope of this manuscript and does not change the overall conclusion drawn from the study.

Fig 4 & 5. Does Erk inhibition rescue the effect in retina vascularization observed in Pdl null mice? There a lot of compound *in vivo* tested that could easily exploited (e.g. refametinib, GDC-0994)

We would like to point out that we already had applied pharmacological inhibition of the Erk1/2 pathway using a MEK inhibitor. We chose this MEK inhibitor for two reasons: 1) there was an *in vivo* study published (Roth et al., Invest Ophthalmol Vis Sci. 2003. PMID: 14638742) in rat showing that it is possible to achieve efficient pErk inhibition in retina with this inhibitor, and 2) we wished to monitor the inhibition of Erk1/2 phosphorylation, since this was the key observation. Erk inhibitors could also be relevant to study, but would not allow us to measure Erk1/2 activation by immunostaining for pT202/pY204 as a way to verify that the inhibitor indeed worked. We regard the data provided as convincing proof of concept.

Referee #2:

The manuscript demonstrates that Paladin can bind to phosphoinositides (PIs) and acts as a phosphatase for PI(4,5)P₂ and PI(3,4,5)P₃. Authors also revealed that Paladin fine-tunes VEGFR2 intracellular trafficking in endothelial cells. In addition, Paladin was found to negatively regulate activation of VEGFR2 and its downstream target ERK1/2 in endothelial cells *in vitro* and *in vivo*. In accordance, Paladin germ-line KO animals showed defects in both developmental and pathological angiogenesis in the mouse retina. These observations are novel and describe new roles for Paladin. Moreover, authors highlighted the relevance of Paladin for VEGFR2 signaling.

However, there are several major and minor issues that need to be addressed prior to publication:

Major issues:

1. The authors show that Paladin is a phosphatase for PIs and that Paladin affects VEGFR2 trafficking. However, importantly, the authors fail to demonstrate that the effect of Paladin on VEGFR2 trafficking is through its phosphatase activity on PIs. To test this, the authors should perform experiments confirming that Paladin regulates VEGFR2 signaling and trafficking via its activity on PIs.

To more strongly link the novel phosphatase activity of Paladin to its effect on VEGFR2 trafficking we have performed a number of new analysis that follows the effect of Paladin loss of function over time after VEGF-A stimulation. In particular, we have analyzed the effect on PI(4,5)P₂ in intact cells after VEGF-A stimulation +/- *PALD1* siRNA, new Figure 2g,h. We observe a sharp accumulation PI(4,5)P₂ in intact cells after *PALD1* knock-down as compared to control cells after 2 min VEGF-A stimulation. Overall, we observe a consistent pattern indicating that Paladin plays an essential role early after VEGF-A stimulation. VEGFR2-Paladin interaction is established at 2 min after VEGF stimulation (Figure 1c,d,e,f). The important role for this co-localization is indicated by faster VEGFR2 internalization (Figure 2c,d), elevated EEA1/VEGFR2 co-localization (Figure 2e,f) and the dramatic increases in PI(4,5)P₂ levels (Figure 2g,h) in response to VEGF-A treatment of Paladin-deficient cells. Taken together, biochemical, cellular and *in vivo* data collectively point to a role for Paladin as a PI(4,5)P₂ phosphate regulating early VEGFR2 trafficking.

In spite of these results, as pointed out by the reviewer, we have not proven that the PI(4,5)P₂-levels *per se* drives this process. Our ambition has been to present the data and

conclusions in a balanced manner and we have been careful to not make the claim that the PI(4,5)P₂-levels directly drives VEGFR2 trafficking. Not even with a new mouse model e.g. expressing kinase inactivated Paladin, could we make that claim. However, we do believe it is fair to state that in the context of previous literature, our data suggests that Paladin regulates VEGFR2 trafficking via its effect on PI(4,5)P₂.

2. Figure 2 A, B, C and lines 161-163 of the main text: "We observed the formation of a Paladin-VEGFR2 complex in response to VEGF-A treatment both in vitro (in primary endothelial cells) and in vivo (in a mouse model)."

This claim is not supported by data. Both in vivo and in vitro data show that VEGF alone is not sufficient to induce complex formation. This only occurs in case of peroxivanadate treatment. Moreover, peroxivanadate alone was sufficient to induce complex formation in vivo. This condition, is a important missing control in Figure 2A and 2B. Input samples should also be included. Along the same line of thought, based on Figure 2D, authors claimed that "Accordingly, super-resolution microscopy analysis confirmed VEGF-A induced co-localization of Paladin and VEGFR2. This indicated that Paladin could be involved in VEGF-A/VEGFR2 signaling." (lines 165-167). Yet, authors do not show conditions with or without VEGF stimulation, and no quantification is provided to support the claim on the levels of co-localization.

We apologize and have now addressed the reviewer's concern. We have replaced the super resolution microscopy with immunostaining analyses (Figure 1b) and added proximity ligation assay (PLA) analysis of VEGFR2 and Paladin interactions showing that VEGF-A induced PLA-complexes occur between VEGFR2 and Paladin in intact cells (Figure 1e,f for controls see Figure EV1). Blots have been complemented and moved to the Figure EV 1. Clarification of the role of peroxyvandate has been included in the text.

3. Figure 2H shows that Paladin KD cells have higher levels of internalised VEGFR2 upon 5-15min of VEGFA stimulation than control cells. However, quantification in Figure 2J and Sup.Figure2F shows that both the number of VEGFR2 vesicles and the number of VEGFR2 vesicles colocalizing with EEA1 at 10min is equivalent to those in Paladin KD cells, whilst one would expect to have higher levels. Can authors comment on these contradictory results? Can authors provide additional confirmatory experiments to clarify this important claim in the manuscript? Moreover, the authors should complement their analysis for VEGFR2 trafficking with additional vesicular markers (Rab5+, Rab7+, Rab4+ or Rab11+ vesicles) in a time course manner to assess the fate of VEGFR2-positive vesicles in control and Paladin KD cells.

We agree that the data as presented in the initial submission was unclear. With improved staining protocol for Paladin, higher resolution imaging and shorter time points for VEGF-A stimulation, it is now clear that EEA1+/VEGFR2+ vesicles increase, especially 2 min after VEGF-stimulation in the absence of Paladin (new Figure 2e,f). We have not tracked the fate of VEGFR2 further in the endosomal compartment, as the data is now concordant with biotinylation and signaling experiments, but rather focused on corroborating the early events with PLA-studies and PI(4,5)P₂ stainings as presented in Figure 1 and 2.

4. In Figure 3, authors show that Paladin KD cells have higher signal transduction in response to VEGFA stimulation in vitro and in vivo. However, data presentation is very puzzling. For instance, Figure3i,j has different merged time points than Figure3f-h. Authors should provide

a consistent way to display time in their analyses. The reviewer suggests that every time point should be displayed individually. In addition, quantification for pVEGFR2 levels in vivo is missing.

We thank the reviewer for this important comment. We have now changed the time points analyzed to be more consistent. However, depending on which signaling molecule you study, they exhibit different peaks of phosphorylation. Moreover, signaling studies *in vivo* are very challenging as the signaling kinetics is fast requiring consistent retrieval and freezing of tissues within minutes after VEGF-A injection in the tail vein, and even for experienced experimentalists, this inevitably introduces variability. Depending on the kinetics of the signaling, we chose to bin some of the time points in the experiments. For different signaling molecules, the peak and duration of signaling is different, and therefore it is reasonable to bin them differently. Please also note that none of the old Figure h-j showed any significant changes, so the time courses are merely there to illustrate the dynamics of the signaling. Consequently, we have moved 3h-j to Figure EV 3. We have updated Figure 3 to also include pVEGFR2 data in vivo, Figure 3f.

5. Figure 4: To understand the extent of the observed rescue upon use of U0126 in Paladin KO animals, it would be important to provide representative images of those that were used for the quantifications represented in these graphs. Also, in figure 4i it is not clear in which condition CyclinD1 stainings were performed (WT or KO?). Representative images for both WT and KO animals should be shown. Moreover, authors could use ERG staining to label endothelial nuclei and thus increase the precision of the quantification (CyclinD1/ERG Double-positive cells), thus excluding confounding contribution of pericytes and microglia.

We have now included representative images for the graphs in the previous Figure 4k,l and placed them in Figure EV 4 f,g. We have updated the Results section and Figure 4 legend to clearly state that the previous Figure 4i (now Figure EV 4d) represents the wildtype condition, to illustrate the CyclinD1 protein localization in the retinal vasculature. In the new Figure 4j there are now representative images of CyclinD1 from wildtype and KO retinas. We could not perform double stain for Erg and Cyclin D1 as both antibodies are from the same species (rabbit). However, we show a high-power image where the reader can see that the CyclinD1 stain is clearly vascular and appears to be endothelial and not in pericytes (Figure EV4d).

6. In general, it would be more informative to show quantification of vascular parameters in Paladin WT and KO retinas in absolute numbers and not as relative to control. Moreover, authors should show all data points (such as in Figure 1B/C), avoiding bars, and privileging scatter dot plots or box and whiskers.

Since developmental angiogenesis in the retina is a very dynamic process and small differences in time point of harvesting the tissue can result in, for example, different vascular outgrowth, we found a normalization to the wild type littermates necessary to pool the data from several litters. However, absolute numbers for vascular parameters were already provided for some of the data in previous Figure EV 4, such as filopodia number, that are less effected by the exact time point of tissue harvest and the data are therefore easier to pool from several litters. We now moved filopodia data to the new Figure 4d. All data are now presented as dot plots.

7. Discussion (line 297-299): "Our data suggest that Paladin is a part of a VEGF-driven negative feedback loop in retinal angiogenesis where VEGF-A upregulates Paladin which acts to dampen VEGFR2 driven signaling and endothelial sprouting"

The authors show that VEGF increases Paladin expression in HUVECS and in the mouse retina upon tail vein injection of VEGF. They do not show the existence of a loop that is interrupted if Paladin is knocked down/ knocked out. Therefore, we believe that in the discussion, the authors should rephrase their sentence to avoid overstating the significance of their findings.

Point well taken; the sentence has been removed.

Minor issues:

1. Figure S1E: It was shown that recombinant Paladin binds specifically to PI(3)P, PI(4)P, and PI(5)P, and PI(3,5)P2. It is somehow surprising not to find Paladin targets [PI(4,5)P2 and PI(3,4,5)P3] as binders. Could the authors explain these results?

A reason for this finding could be that Paladin binds to phosphoinositides other than the substrates. For example, it is known that PTEN binds to PI(3)P, which mediates its localization to endosomes, but dephosphorylate PI(3,4,5)P3 (Naguib, Bencze et al. Mol Cell. 2015). Importantly, Naguib et al. show that the PIP array does not indicate binding of PTEN to PI(3,4,5)P3, even though it is the substrate. A reason for this could perhaps be that the catalytic site mediates a more transient, low affinity interaction. However, as we don't follow up the potential interaction of Paladin and PIPs with further experiments, it is reasonable to omit the data to avoid confusion.

2. Line 172-173: "However, the receptor was degraded at the same rate as control-treated cells after VEGF-A stimulation (Figure 2e, Suppl Figure 2b)." It is unclear how the authors concluded about degradation rates. Could the authors clarify?

The slope of the curve for the VEGFR2 protein 0-60 min after VEGF stimulation in the previous Suppl Figure 2b (now Figure EV2b) is similar for Paladin knock-down and ctrl treated cells, suggesting that VEGFR2 was degraded at the same rate in the presence and absence of Paladin. However, as we have not formally determined the degradation rate, we have changed the wording to: " However, the receptor was degraded similarly over time after VEGF-A stimulation when comparing *PALD1* siRNA and control treated cells (Figure 2a, Figure EV 2b)."

3. It would be interesting to provide images of P5 retinas images with higher magnification of tip cells from lacZ-stained Paladin het animals, similar to Figure 5a/c.

Indeed, the Paladin LacZ reporter localizes nicely to tip cells in the sprouting retina. We have published this in Wallgard et al Dev Dyn 2012 Figure 6i and 8c

4. Figure 4: To strengthen their findings, it would be relevant to show the expression pattern of VEGFR2 and pVEGFR2 in the wt and Pald1 KO retinas.

We believe that the analysis suggested would not bring us further in the understanding of Paladin's role on VEGFR2 signaling and trafficking. Moreover, pVEGFR2 detection is technically very challenging, if not impossible to perform *in vivo* in most tissues, most likely since the phosphorylated receptor pool is small and very unstable. Our *in vitro* and *in vivo* signaling studies suggest that we should expect a shift in the rate of internalization of pVEGFR2 rather than any broad changes in the overall pVEGFR2 levels.

5. Line 144-145: "Rab4, -7, and -11, markers of fast recycling, slow recycling and late endosomes, respectively". This statement is incorrect. Rab7 marks late endosomes and while Rab11 marks slow recycling endosomes. The sentence should be corrected.

The Rab data has now been removed as discussed above, Q3.

6. Line 147-148: "Super-resolution microscopy revealed that one-quarter of the Rab4- or Rab11-positive structures were also positive for Paladin". The quantification for the number of vesicles that are double positive for Paladin/Rab4 and for Paladin/Rab11 should be showed in a graph.

We have now provided extensive quantification of Paladin in relation to VEGFR2 and EEA1 and focused on these interactions and removed the Rab stainings as this is no longer the focus of the paper, as also discussed above

7. Line 170-171: "siRNA-mediated knockdown of PALD1 in HDMEC resulted in a 35-51% increase of the total basal VEGFR2 pool". It is unclear what technique was used to obtain this result. It is not stated in the main text nor in the figure legend. If the technique used was Western Blot, it would be important to show the image with the bands that allowed to make this quantification.

We used western blot to quantify the Paladin levels and we have modified the text to state that clearly, see Figure 2a,c and Figure EV 2a.

8. Line 237-238: For clarity, the authors should clarify in the main text which animal model was used. From the main text it seems that an inducible endothelial specific knock out animal was used. But from reading the materials and methods, that does not seem to have been the case.

The text has been updated to state the use of a constitutive Pald1 knock-out mouse.

9. Line 342: this reference "Lanahan, 2010" has not been included in the bibliography section.

This has now been corrected.

10. Line 346: "We also observed increased pTyr1173 phosphorylation in HDMEC". For clarity, the authors should rephrase this sentence to be more accurate, namely increased in which condition as compared with what.

Text has been corrected.

11. Figure 1D, right panel: the expression pattern of paladin and PI4P seem to be heterogeneous across the cell population and perhaps inversely correlated. Is that the case? Could the authors show separate panels for each colour? Could the authors comment on this?

The reviewer is correct that there was some variation in the staining intensity, but the images presented exaggerated that difference. Staining homogeneity has been improved. However, the data has been removed as the focus of Figure 1 has shifted (see above).

12. Figure 4K and L: Could the authors clarify why two regimes of U0126 administration were used?

The maximum inhibition of pERK was observed 2-4.5 h after dosing. We used two different regimens as the filopodia that characterize the tip cells are highly dynamic structures. We reasoned that we should observe a potential phenotype with 4h treatment which was also practically feasible. Migration of endothelial cells in the retinal plexus is a slower process and we had to wait 24 h before we could expect to see an effect and therefore, we spaced the injection interval accordingly. A text describing this rationale has been included in M&M.

13. Line 271: "VEGF-A induced production of the Paladin protein in endothelial cells *in vitro* and in the retinal vasculature *in vivo*, as indicated by LacZ reporter expression (Figure 5b,c and Suppl Figure 5a)." Authors should clarify this text as it suggests that quantifications were performed on LacZ reporter, yet, Figure 5B seems to be WB from bands showed in Sup Figure 5a.

The reviewer is correct, we have now changed it to:

"Indeed, VEGF-A, but not other endothelial cell growth factors such as fibroblast growth factor-2 (FGF2) or stromal derived factor 1a (SDF1a) significantly induced expression of Paladin in endothelial cells *in vitro* (Figure 5b, Figure EV 5a). Moreover, Paladin was induced in the retinal vasculature *in vivo*, as indicated by LacZ reporter expression (Figure 5c)."

14. Figure S1D: it is not very clear what portion of the cell this image is reporting. The authors should provide an additional image with the zoom out of this cell with the location of the zoom in marked.

Previous Figure S1D has been replaced with the new Figure 1b to better show the overview and zoomed in parts.

Dear Mats,

Thank you for submitting your revised manuscript. It has now been seen by both of the original referees.

As you can see, the referees find that the study is significantly improved during revision and recommend publication. Before I can accept the manuscript, I need you to address some minor points below:

- Please address the remaining minor concerns of referee #2.
- As per our guidelines, please add a 'Data Availability Section', where you state that no data were deposited in a public database.
- We think your manuscript fits better to the format of a 'Scientific Report', and therefore the Results and the Discussion sections should be combined. Please see <https://www.embopress.org/page/journal/14693178/authorguide#researcharticleguide> for more details.
- We notice that there are references in the Materials & Methods section as well. Please move them to the main reference list.
- All articles published beginning 1 July 2020, the EMBO Reports reference style changed to the Harvard style for all article types. Details and examples are provided at <https://www.embopress.org/page/journal/14693178/authorguide#referencesformat>. Please update the reference style accordingly.
- We notice that the funding information is not complete in the manuscript submission system.
- We realized that Figures EV2-EV5 are currently not called out in the text.
- We note that the graph of Figure EV1C is missing scale bars.
- We realized that Appendix Figures 2-9 are actually source data. Please upload them as source data by separating them one file per figure and remove the callouts from the text.

Thank you again for giving us to consider your manuscript for EMBO Reports, I look forward to your minor revision.

Kind regards,

Deniz

--

Deniz Senyilmaz Tiebe, PhD
Editor
EMBO Reports

Referee #1:

The authors have well clarified my doubts and concerns

Referee #2:

The reviewer congratulates the authors for the excellent revision of the manuscript, and for having put a strong effort with numerous new experiments. This is especially appreciated given the COVID-19 related restrictions.

The authors have satisfactorily answer to all the points raised, and I support publication at this stage.

Two small details:

In Figure B2, microglobulin is misspelled.

Figure EV4f - the arrow in the upper right panel is misplaced. The arrowhead should be located in the edge of the sprouting front (as it is in the remaining panels of this image).

Point-by-point letter EMBOR-2020-50218V3

Dear Dr Senyilmaz,

We have now addressed all the comments from reviewer #2 and your technical comments.

Please find a point-by-point response below.

- Please address the remaining minor concerns of referee #2.

Done. However, reviewer #2 misunderstood the arrow in Figure EV4f as it is supposed to be the same length in wild-type and knockout. We have updated the Figure legend to make it clear.

- As per our guidelines, please add a 'Data Availability Section', where you state that no data were deposited in a public database.

We have added "Data availability section in the MS"

- We think your manuscript fits better to the format of a 'Scientific Report', and therefore the Results and the Discussion sections should be combined. Please see <https://www.embopress.org/page/journal/14693178/authorguide#researcharticleguide> for more details.

We have combined Results and Discussion and shortened the text to comply with the format.

- We notice that there are references in the Materials & Methods section as well. Please move them to the main reference list.

References have been removed from M&M and merged into one section of References.

- All articles published beginning 1 July 2020, the EMBO Reports reference style changed to the Harvard style for all article types. Details and examples are provided at <https://www.embopress.org/page/journal/14693178/authorguide#referencesformat>. Please update the reference style accordingly.

We have updated the reference style.

- We notice that the funding information is not complete in the manuscript submission system.

We have updated funding information in your system.

- We realized that Figures EV2-EV5 are currently not called out in the text.

This was a misunderstanding as we discussed per e-mail. They are called out in the text.

- We note that the graph of Figure EV1C is missing scale bars.

Error bars have been added.

- We realized that Appendix Figures 2-9 are actually source data. Please upload them as source data by separating them one file per figure and remove the callouts from the text. Appendix has been updated and source data has been uploaded

Thank you again for giving us to consider your manuscript for EMBO Reports, I look forward to your minor revision.

Kind regards,

Deniz

--

Deniz Senyilmaz Tiebe, PhD
Editor
EMBO Reports

Referee #1:

The authors have well clarified my doubts and concerns

Referee #2:

The reviewer congratulates the authors for the excellent revision of the manuscript, and for having put a strong effort with numerous new experiments. This is especially appreciated given the COVID-19 related restrictions.

The authors have satisfactorily answer to all the points raised, and I support publication at this stage.

Two small details:

In Figure B2, microglobulin is misspelled.

Figure EV4f - the arrow in the upper right panel is misplaced. The arrowhead should be located in the edge of the sprouting front (as it is in the remaining panels of this image).

Dear Mats,

Thank you for submitting your revised manuscript. I have now looked at everything and all is fine. Therefore I am very pleased to accept your manuscript for publication in EMBO Reports.

Congratulations on a nice study!

Kind regards,

Deniz

--

Deniz Senyilmaz Tiebe, PhD
Editor
EMBO Reports

--

At the end of this email I include important information about how to proceed. Please ensure that you take the time to read the information and complete and return the necessary forms to allow us to publish your manuscript as quickly as possible.

As part of the EMBO publication's Transparent Editorial Process, EMBO reports publishes online a Review Process File to accompany accepted manuscripts. As you are aware, this File will be published in conjunction with your paper and will include the referee reports, your point-by-point response and all pertinent correspondence relating to the manuscript.

If you do NOT want this File to be published, please inform the editorial office within 2 days, if you have not done so already, otherwise the File will be published by default [contact: emboreports@embo.org]. If you do opt out, the Review Process File link will point to the following statement: "No Review Process File is available with this article, as the authors have chosen not to make the review process public in this case."

Should you be planning a Press Release on your article, please get in contact with emboreports@wiley.com as early as possible, in order to coordinate publication and release dates.

Thank you again for your contribution to EMBO reports and congratulations on a successful publication. Please consider us again in the future for your most exciting work.

THINGS TO DO NOW:

You will receive proofs by e-mail approximately 2-3 weeks after all relevant files have been sent to our Production Office; you should return your corrections within 2 days of receiving the proofs.

Please inform us if there is likely to be any difficulty in reaching you at the above address at that time. Failure to meet our deadlines may result in a delay of publication, or publication without your corrections.

All further communications concerning your paper should quote reference number EMBOR-2020-50218V3 and be addressed to emboreports@wiley.com.

Should you be planning a Press Release on your article, please get in contact with emboreports@wiley.com as early as possible, in order to coordinate publication and release dates.

EMBO REPORTS

Corresponding Author Name: Mats Hellström
Manuscript Number: EMBOR-2020-50218V1